# CLUSCOMP: A SIMPLE PARADIGM FOR MODEL COMPRESSION AND EFFICIENT FINETUNING

## ABSTRACT

As large language models (LLMs) continue to scale, model compression becomes increasingly important for enabling edge deployment and ensuring accessibility to users with limited resources. Weight-only quantization is a key technique for model compression, allowing for a substantial reduction in model size while preserving performance. However, as bit-width decreases, the performance of quantized LLMs tends to degrade significantly. Additionally, due to the non-differentiable operation in quantization, standard finetuning on quantized LLMs is unsupported, and alternative finetuning approaches often fail to match the effectiveness of full finetuning. In this paper, we introduce *ClusComp*, a novel and simple model compression paradigm. ClusComp first clusters the weight matrices to generate codebooks, and then tunes these codebooks block-by-block to reconstruct intermediate activations. Despite its simplicity, ClusComp (1) consistently achieves better performance in 2-4 bit precision; (2) pushes the compression limit to the 1-bit level, and outperforms existing ultra-low-bit methods with limited finetuning steps; (3) facilitates seamless and efficient finetuning, surpasses existing quantization-based or memory-efficient finetuning methods, and even rivals full finetuning of the FP16 model. Notably, these procedures can be executed on a single NVIDIA A6000-48GB GPU for LLMs with as many as 70B parameters.

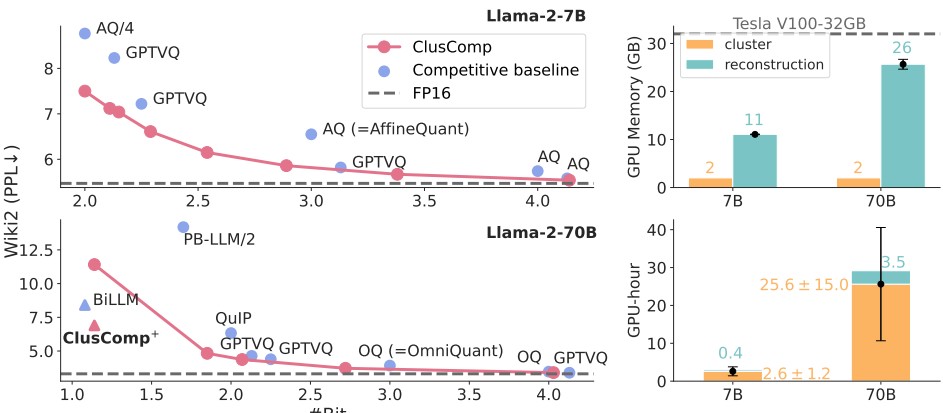

Figure 1: Compression quality and efficiency of ClusComp, consisting of a sequential clustering and reconstruction. Methods in triangle use more number of calibration samples. Some results are divided by a factor for better visualization. E.g. "AQ/4" indicates that the perplexity is divided by 4.

## 1 INTRODUCTION

Large language models (LLMs) have garnered significant acclaim and success across various domains and applications (Touvron et al., 2023a; Brown et al., 2020; Raffel et al., 2020b). With ongoing advancements, the scope and complexity of released LLMs have witnessed exponential growth, with some LLMs encompassing >50B parameters (Dubey et al., 2024; Zhang et al., 2022; Scao et al., 2022). This remarkable upscaling introduces considerable challenges, particularly when

deploying these models or granting their accessibility to users with constrained resources. To address these challenges, weight-only post-training quantization (PTQ) has emerged as a promising approach, effectively compressing LLMs to a lower bit while preserving the FP16 performance.

PTQ methods can generally be classified into three categories: statistic-based, gradient-based, and codebook-based approaches. Statistic-based methods (Dettmers et al., 2024; Lin et al., 2024; Frantar et al., 2022) determine the quantization grid based on the distribution of the weight values, whereas gradient-based methods (Shao et al., 2024; Ma et al., 2024) optimize the quantization grid with some calibration samples. Codebook-based methods (Egiazarian et al., 2024; van Baalen et al., 2024; Kim et al., 2024; Park et al., 2024) cluster similar weight elements to the shared quantized centroids, employing non-uniform quantization and pushing the limits to extremely low bit levels. However, these methods continue to struggle with low-bit quantization and the presence of outliers, leading to significant performance degradation, especially in models like Llama-3 (Dubey et al., 2024), which exhibit a large number of outliers in their weight matrices (Huang et al., 2024c).

Another challenge PTQs encounter is their limited support for finetuning, which is crucial for adapting LLMs to various downstream tasks. Finetuning LLMs is computationally expensive due to their large scale and the need to cache activations and store optimizer states. PTQ, which compresses LLMs, appears to be a promising approach for finetuning as it reduces memory requirements for loading these LLMs. However, most quantization techniques use a round-to-nearest operation, which does not support gradient back-propagation. Typically, parameter-efficient methods (Dettmers et al., 2023; Li et al., 2024c; Liao & Monz, 2024a) are employed to train the added parameters while keeping the quantized LLMs frozen, bypassing this limitation. Nonetheless, this finetuning approach presents two major drawbacks: (1) Freezing the quantized LLMs prevents further reduction of quantization errors during finetuning; (2) The low-rank nature of most parameter-efficient methods restricts their expressiveness (Biderman et al., 2024; Liao & Monz, 2024b).

In this paper, we propose a simple while effective paradigm that mainly applies **Clus**tering to **Comp**ress LLMs, referred to as *ClusComp*. Additionally, ClusComp can function as a parameter and memory-efficient finetuning method. Our preliminary experiments reveal that open-source LLMs are increasingly difficult to quantize, primarily due to the growing frequency of outliers in their weight matrices (§3.1). Based on this observation, we propose using clustering instead of quantization to compress LLMs, retaining all values in FP16 format to circumvent issues arising from outlier quantization (§3.2.1). To further reduce compression errors, we minimize block-wise output discrepancies between the compressed and uncompressed blocks, using a limited set of calibration samples (§3.2.3). Since all parameters remain in FP16 after compression, ClusComp fully supports standard neural network training. By incorporating an inexpensive, end-to-end recovery finetuning step, we can push compression rates to the 1-bit level. Additionally, ClusComp allows for finetuning compressed LLMs on various downstream tasks (§3.2.4).

We begin by evaluating the effectiveness of ClusComp in the context of model compression across 2 language modeling tasks and 6 zero-shot reasoning tasks. ClusComp consistently surpasses various baselines at 2-4 levels, even achieving a perplexity of <13 at the 2-bit level on WikiText2 (Merity et al., 2017) for all LLMs (§4.1). Following recovery finetuning, ClusComp's performance at 2-bit and 1-bit levels approaches that of the FP16 model, with an accuracy of 57.8 vs 68.6 for the 2-bit Llama-3-8B and 51.4 vs 75.4 for the 1-bit Llama-3-70B (§4.2). Additionally, ClusComp demonstrates its utility as a parameter-efficient ($< 1\%$) and memory-efficient (42GB for Llama-3-70B) finetuning method, outperforming quantization-based and memory-efficient finetuning approaches, while matching the performance of full finetuning (§4.3).

## 2 RELATED WORKS

### 2.1 MODEL COMPRESSION

Quantization and pruning are two typical and effective methods for model compression.

**Quantization** refers to the process of converting floating-point values into discrete levels, thereby reducing the bit-width required and minimizing memory consumption during model loading. Taking

the symmetric uniform quantization as an example, a weight matrix $\boldsymbol{W}$ is quantized as follows:

$$\boldsymbol{W}_{\mathrm{q}} = \mathrm{clamp}(\lfloor \frac{\boldsymbol{W}}{s} \rceil, -2^{b-1}, 2^{b-1} - 1) \quad \text{with} \quad s = \frac{\max(|\boldsymbol{W}_{\min}|, |\boldsymbol{W}_{\max}|)}{2^b - 1} \tag{1}$$

where $b$ denotes the bit-width, $s$ is the scale factor, and $\lfloor \rceil$ represents the round-to-nearest (RTN) operation. Since the quantization grid is uniform, its effectiveness is contingent on the distribution of the weight values. In cases where the weight matrix contains a significant number of outliers or is quantized to lower bit-widths, the resulting quantization error may be substantial.

**Post-training quantization** (PTQ) methods, such as GPTQ (Frantar et al., 2022), AWQ (Lin et al., 2024), and OmniQuant (Shao et al., 2024), apply quantization to a model after training with minimal computational resources. However, these approaches, which rely on uniform quantization, are significantly impacted by the presence of outliers in the weight matrices. Recent methods (Dettmers et al., 2024; Yuan et al., 2024; Huang et al., 2024a) address this challenge by retaining salient weights in FP16 format, thereby maintaining strong performance at lower bit widths. Nonetheless, these mixed-precision quantization techniques require specially optimized CUDA kernels to either enhance or preserve inference speed. Closely related to our proposed method, ClusComp, are works such as GPTVQ (van Baalen et al., 2024), QuIP# (Tseng et al., 2024) and SqueezeLLM (Kim et al., 2024) which implement quantized codebooks for non-uniform quantization, achieving state-of-the-art performance for ultra-low-bit quantization. ClusComp, however, differs in two significant ways: (1) The codebook in ClusComp is stored in FP16, offering additional advantages for subsequent recovery training and finetuning; (2) While other methods face limitations similar to those in VAE-like approaches (Kingma & Welling, 2014), where large and high-dimensional codebooks are infeasible due to mode collapse, ClusComp circumvents this issue. Our fixed-code design allows us to utilize a codebook size of $2^{16}$ in 4-16D without encountering such difficulty.

**Pruning** is a widely used model compression technique that removes redundant weights or structures from the model (Sun et al., 2024a; Xia et al., 2024; Frantar & Alistarh, 2023; Liao et al., 2023). It often leads to significant degradation as sparsity increases, and generally yields inferior results at equivalent compression rates compared to quantization. Nevertheless, pruning offers an advantage for training, as all parameters remain in high precision, allowing for seamless integration with continuous pretraining or finetuning. Similarly, ClusComp retains high-precision parameters, and naturally supports standard finetuning.

## 2.2 KNOWLEDGE DISTILLATION

Knowledge distillation is a technique used to enhance the performance of smaller models by transferring knowledge from larger, more complex models (Hinton et al., 2015). Most state-of-the-art quantization methods leverage either block-wise or model-wise distillation. Block-wise distillation (as employed in OmniQuant, GPTVQ, AQLM and QuIP#) focuses on minimizing errors between the FP16 and quantized models on a block-by-block basis. This approach is more memory-efficient than model-wise distillation, as it requires loading only two blocks into the GPU at the same time. In contrast, model-wise distillation (used in QuIP# and LLM-QAT (Liu et al., 2024)) minimizes the error across the entire model output, necessitating the loading of at least one full FP16 model into the GPU. ClusComp adopts block-wise distillation, significantly reducing GPU memory requirements compared to loading an FP16 model. As demonstrated in Figure 1, ClusComp consumes only 26GB memory for a 70B LLM, which would otherwise require 140GB in FP16 for loading, making our technique more accessible to users with limited computational resources.

## 2.3 FINETUNE QUANTIZED MODEL

Finetuning is crucial for adapting LLMs to various domains and applications. Quantization, which reduces model size, is theoretically more conducive to finetuning. However, directly finetuning a quantized model is not a standard approach, as RTN does not support gradient back-propagation. Finetuning using a straight-through estimator (STE) (Bengio et al., 2013) is relatively under-explored and may lead to catastrophic forgetting (Malinovskii et al., 2024). Previous works (Xu et al., 2024a; Liao & Monz, 2024a; Dettmers et al., 2023) propose freezing the quantized model while updating newly added LoRAs (Hu et al., 2022). However, these approaches suffer from two key limitations: (1) Freezing the quantized model prevents the mitigation of quantization errors during finetuning. Moreover, not all quantization methods are suitable for finetuning. Popular

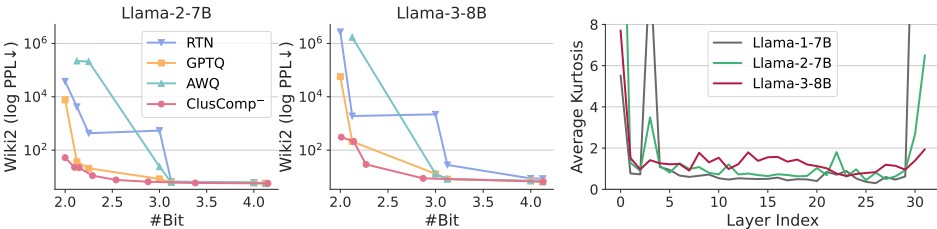

Figure 2: The Llama series becomes increasingly difficult to quantize. **Left** & **Middle**: From Llama-2 to Llama-3, all methods show increasing difficulty in quantization at lower bit levels. **Right**: From Llama-1 to Llama-3, the average kurtosis on weights of most layers is increasing.

techniques like GPTQ and QLoRA, while widely used, exhibit significant quantization errors below 4-bit. (2) The expressiveness of LoRA is constrained by its bottleneck design (Biderman et al., 2024; Liao & Monz, 2024b). In contrast, ClusComp, where all parameters are maintained in high precision, inherently supports seamless finetuning. Additionally, updating the codebook in ClusComp results in modifying all parameters in the weight matrices, even offering superior performance compared to full finetuning while maintaining a similar number of trainable parameters as LoRA.

## 3 METHOD

### 3.1 PILOT STUDY

Before introducing ClusComp, we present a key observation from our experiments on different Llama series. As depicted in Figure 2 (Left & Middle), when reducing the bit-width, the performance of various quantization methods (RTN, GPTQ, and AWQ) follows a similar trend: Llama-3 (Dubey et al., 2024) proves more challenging to quantize than Llama-2 (Touvron et al., 2023b).

We hypothesize that the increased difficulty arises from a higher frequency of outliers in the linear layers of Llama-3. Since these quantization methods rely on uniform quantization, they are particularly sensitive to outliers in the weight matrices. To test this hypothesis, we analyzed the kurtosis of the weight matrices—a well-established metric for identifying the presence of outliers (Bondarenko et al., 2023). As shown in Figure 2 (Right), we have two key observations: (1) All models exhibit higher kurtosis at the beginning and end of the model; (2) From Llama-1 to Llama-3, the kurtosis increases in most layers, indicating a rise in the frequency of outliers in the linear layers. This trend provides a potential explanation for our quantization difficulties. It also implies that the future Llama series might be even more difficult for quantization.[1]

*Given that the quantization performance is impacted by outliers, could an alternative approach for model compression involve storing all weight values in FP16 instead of applying quantization?*

### 3.2 CLUSCOMP

The first idea that comes to our mind is clustering, where similar weight values are represented by a single identified value. This method enables model compression while preserving all weight values in FP16 format. In this section, we introduce three variants of ClusComp that primarily utilize clustering for compressing LLMs: ClusComp$^-$, which applies clustering alone; ClusComp, which enhances ClusComp$^-$ with block-wise error minimization; and ClusComp$^+$, which further improves the compressed LLMs through next-token prediction training based on ClusComp.

#### 3.2.1 CLUSTERING

Consider a weight matrix $\boldsymbol{W} \in \mathbb{R}^{d_{\text{in}} \times d_{\text{out}}}$, direct clustering along either dimension of $\boldsymbol{W}$ is suboptimal as it leads to a significant reconstruction error, particularly due to the large values of $d_{\text{in}}$ and $d_{\text{out}}$ in LLMs. To mitigate this issue, we reshape $\boldsymbol{W}$ into a set of lower-dimensional vectors, denoted as

---

[1]We present the kurtosis for different types of layers and a promising quantization idea in Figure C.1.

Table 1: Bits for $\boldsymbol{W}$ with $d_{\text{in}}, d_{\text{out}} = 4096$ (16.78M).

| Setting | #Params. for codes | #Params. for codebook | $\bar{b}$ |
|---------|--------------------|-----------------------|-----------|
| g4n65500 | 4.19M | 0.26M (1.55%) | 4.25 |
| g6n65500 | 2.80M | 0.39M (2.32%) | 3.04 |
| g9n65500 | 1.86M | 0.59M (3.52%) | 2.34 |

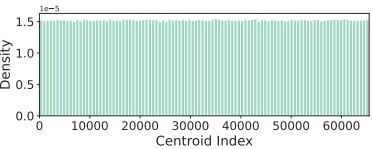

Figure 3: Histogram of the codes.

$\boldsymbol{W}' = \{\boldsymbol{w}_1, \boldsymbol{w}_2, \ldots, \boldsymbol{w}_k\}$, where each $\boldsymbol{w}_i \in \mathbb{R}^g$ and $k = \frac{d_{\text{in}} \cdot d_{\text{out}}}{g}$.[2] The goal is to partition $\boldsymbol{W}'$ into $n$ clusters $\{C_1, C_2, \ldots, C_n\}$ by solving the following optimization problem:

$$\arg\min_{\{C_1, C_2, \ldots, C_n\}} \sum_{j=1}^{n} \sum_{\boldsymbol{w}_i \in C_j} ||\boldsymbol{w}_i - \boldsymbol{c}_j||^2 \tag{2}$$

where $\boldsymbol{c}_j \in \mathbb{R}^g$ denotes the centroid of cluster $C_j$. This clustering problem is well-established in the machine learning literature and can be iteratively addressed using K-means (Lloyd, 1982) with the Expectation-Maximization (EM) algorithm:

- **E-step**: Each vector $\boldsymbol{w}_i$ is assigned to the cluster whose centroid $\boldsymbol{c}_j$ minimizes the Euclidean distance, i.e., $C_j^{(t)} = \{\boldsymbol{w}_i : ||\boldsymbol{w}_i - \boldsymbol{c}_j^{(t)}||^2 \leq ||\boldsymbol{w}_i - \boldsymbol{c}_l^{(t)}||^2 \quad \forall l\}$.
- **M-step**: The centroid of each cluster is updated as the mean of the vectors assigned to that cluster, i.e., $\boldsymbol{c}_j^{(t+1)} = (\sum_{\boldsymbol{w}_i \in C_j^{(t)}} \boldsymbol{w}_i)/|C_j^{(t)}|$.

Upon completion, two key elements are obtained for each weight matrix: (1) a codebook $\boldsymbol{C} = \{\boldsymbol{c}_1, \boldsymbol{c}_2, \ldots, \boldsymbol{c}_n\} \in \mathbb{R}^{g \times n}$ that contains all centroids, and (2) a set of codes $\boldsymbol{q} = \{q_1, q_2, \ldots, q_k\} \in \{1, 2, \ldots, n\}^k$ that records the assignment of each vector $\boldsymbol{w}_i$ to the closest centroid, where $q_i = q(\boldsymbol{w}_i) = j$ if $\boldsymbol{c}_j = \arg\min_{\boldsymbol{c}_l \in \boldsymbol{C}} ||\boldsymbol{w}_i - \boldsymbol{c}_l||^2$. Using the codes $\boldsymbol{q}$ and the codebook $\boldsymbol{C}$, the weight matrix $\boldsymbol{W}'$ can be reconstructed as $\hat{\boldsymbol{W}}' = \{\boldsymbol{c}_{q_1}, \boldsymbol{c}_{q_2}, \ldots, \boldsymbol{c}_{q_k}\}$. In PyTorch (Paszke et al., 2017), the linear layer is adapted in Listing C.1.

*Remark*: ClusComp, when applied solely with clustering, is referred to as ClusComp⁻. In this configuration, only the weight matrices are utilized, leading to substantial memory efficiency, with a mere 2GB memory consumption on 1 GPU as shown in Figure 1. Moreover, this process can be considerably accelerated with more GPUs, as the clustering of different matrices is independent.

### 3.2.2 ESTIMATE MODEL SIZE

After clustering, it is sufficient to store the codes $\boldsymbol{q} \in \{1, 2, \ldots, n\}^k$ and the codebook $\boldsymbol{C} \in \mathbb{R}^{g \times n}$. Unlike prior works (van Baalen et al., 2024; Egiazarian et al., 2024; Tseng et al., 2024), we don't quantize the codebook; instead, we store it in FP16 format. The bit-width required for the codes depends on the range of $n$. To maintain efficiency, we set $n < 2^{16}$ and use unsigned 16-bit integers to represent the codes. Thus, the average bits-per-parameter can be calculated as:

$$\bar{b} = \frac{\text{size in bits}}{\text{number of parameters}} = \frac{16 \cdot k + 16 \cdot g \cdot n}{d_{\text{in}} \cdot d_{\text{out}}} = \frac{16}{g} + \frac{16 \cdot g \cdot n}{d_{\text{in}} \cdot d_{\text{out}}} \tag{3}$$

In the right-most of Equation (3), the first term corresponds to the bit-width allocated to the codes, and the second term corresponds to the bit-width of the codebook. As an example, for a linear layer $\boldsymbol{W}$ with $d_{\text{in}} = d_{\text{out}} = 4096$, clustering with $g = 4$ and $n = 2^{16} - 1$ results in $\bar{b} \approx 4 + 0.25 = 4.25$. This demonstrates that the majority of the bit-width is allocated to the codes, which is a primary reason for constraining $n < 2^{16}$, so we can use 16 instead of 32-bit integers to represent the code. Further reducing $n$ to a smaller range leads to fewer centroids, which in turn increases reconstruction error. More settings can be found in Table 1 and C.1.

*Remark*: While we express the model size in terms of bits-per-parameter, it is important to note that no quantization is applied in ClusComp. Instead, we reduce the number of parameters in $\boldsymbol{W}$ from

---

[2]In cases where the dimensions are not divisible, zero-padding is applied to $\boldsymbol{W}$. In PyTorch, the matrix is reshaped as $\boldsymbol{W}' = \boldsymbol{W}$.transpose(1, 0).view(-1, g), where transposing $\boldsymbol{W}$ offers slightly better performance.

$d_\text{in} \cdot d_\text{out}$ to $k + g \cdot n$. Utilizing bits-per-parameter allows for a direct comparison between ClusComp and quantization-based methods. Surprisingly, ClusComp$^-$, which only incorporates the clustering step without employing any calibration data, already surpasses RTN, GPTQ and AWQ, as illustrated in Figure 2 (Left & Middle). These results highlight its effectiveness in model compression.

### 3.2.3 BLOCK-WISE ERROR MINIMIZATION

Block-wise error minimization (block-wise reconstruction or knowledge distillation) has emerged as a standard, efficient and effective approach to reducing quantization error (Egiazarian et al., 2024; Tseng et al., 2024; van Baalen et al., 2024; Shao et al., 2024; Liao & Monz, 2024a). To further mitigate the compression error caused by clustering, we incorporate block-wise error minimization into ClusComp$^-$ using a limited set of calibration samples, expressed as:

$$\arg \min_{Cs} ||\mathcal{F}(\boldsymbol{W}s, \boldsymbol{X}) - \mathcal{F}(\boldsymbol{C}s, \boldsymbol{q}s, \boldsymbol{X}')|| \tag{4}$$

Here, $\mathcal{F}$ denotes a Transformer block (Vaswani et al., 2017), $\boldsymbol{W}s$ represent the weight matrices in the uncompressed block, and $\boldsymbol{C}s$ and $\boldsymbol{q}s$ denote the codebooks and codes in the compressed block. $\boldsymbol{X}$ refers to the input of the uncompressed block, which is also the output from the previous uncompressed block, while $\boldsymbol{X}'$ is the input to the compressed block, originating from the output of the preceding compressed block. For the first block, we have $\boldsymbol{X} = \boldsymbol{X}'$. Block-wise error minimization is memory-efficient as it only requires loading two blocks into the GPU simultaneously.

*Remark*: In Equation (4), we only train the codebook $\boldsymbol{C}s$ while keeping the codes $\boldsymbol{q}s$ fixed as indices. This design offers two key advantages: (1) It enhances data efficiency. As illustrated in Table 1, the majority of parameters are represented by the codes. Training both the codebooks and codes with a limited number of calibration samples (128) leads to overfitting; (2) More importantly, training the codes with a large number of centroids ($2^{16}$) can result in mode collapse (Sun et al., 2024b; Kingma & Welling, 2014). Since the codes already exhibit a uniform distribution after clustering (see Figure 3), keeping the codes fixed indicates that all centroids in the codebook can be trained uniformly. Such a code-fixed design is also applied to the following recovery and finetuning step. Combining both clustering and block-wise error minimization steps, we term this method ClusComp.

### 3.2.4 RECOVERY AND FINETUNING

We present the adapted linear layer for ClusComp in Listing C.1, which can be seamlessly integrated as a replacement for the original linear layer in LLMs. As the codebook is represented in FP16, this new layer inherently supports training without requiring additional tricks, like STE.

**Recovery training.** The compressed LLMs can be further trained by predicting the next token to recover information lost due to compression. This is achieved by finetuning the codebook parameters. This form of training is memory-efficient in two distinct ways: (1) Since the LLM is already compressed, loading it onto the GPU consumes less memory compared to the FP16 version; (2) As illustrated in Table 1, the parameters in the codebook account for $< 5\%$ of the total parameters in the FP16 version, making the training both parameter-efficient and memory-efficient (with the optimizer states being smaller). We refer to ClusComp with recovery training as ClusComp$^+$.

**Finetuning.** Like recovery training, finetuning the compressed LLM on downstream tasks can also be performed efficiently. Unlike QLoRA (Dettmers et al., 2023), which freezes the quantized LLM and trains only the LoRA (Hu et al., 2022) modules, finetuning the codebook alone eliminates the need for this additional constraint. This approach offers two key advantages over QLoRA: (1) Freezing the quantized LLM prevents mitigation of quantization errors, whereas finetuning the codebook can further address compression errors for downstream tasks; (2) The low-rank bottleneck of LoRA limits its expressiveness (Biderman et al., 2024). In contrast, finetuning the codebook is analogous to adapting the entire high-rank weight matrix, providing greater flexibility and expressiveness.

## 4 EXPERIMENTS

### 4.1 COMPRESSION RESULTS

**LLMs and evaluation.** We evaluate ClusComp on widely adopted LLM families: Llama-1-7B, Llama-2-7B/13B/70B and Llama-3-8B/70B (Touvron et al., 2023a;b; Dubey et al., 2024). We measure the performance of compressed LLMs on zero-shot and language modeling tasks. For zero-shot

Table 2: The perplexity of Llama series on WikiText2 and C4. Only the competitive baselines are shown here for a compact representation. Refer to Table C.1 and C.2 for all results and settings.

| Method | #Bit | Wiki2 (PPL↓) | | | | | | C4 (PPL↓) | | | | | |
|---|---|---|---|---|---|---|---|---|---|---|---|---|---|
| | | 1-7B | 2-7B | 2-13B | 2-70B | 3-8B | 3-70B | 1-7B | 2-7B | 2-13B | 2-70B | 3-8B | 3-70B |
| - | 16.00 | 5.68 | 5.47 | 4.88 | 3.31 | 6.12 | 2.90 | 7.08 | 6.97 | 6.46 | 5.52 | 9.20 | 5.87 |
| GPTQ | 4.13 | 5.85 | 5.61 | 4.98 | 3.42 | 6.50 | 3.30 | 7.21 | 7.12 | 6.56 | **5.58** | 10.40 | **6.94** |
| AffineQuant | 4.13 | 5.77 | 5.58 | 4.95 | - | - | - | 7.20 | 7.12 | 6.56 | - | - | - |
| GPTVQ | 4.13 | - | 5.68 | 4.97 | 3.39 | - | - | - | - | - | - | - | - |
| OmniQuant | 4.16 | 5.77 | 5.58 | 4.95 | 3.40 | - | - | 7.21 | 7.12 | 6.56 | 5.58 | - | - |
| ClusComp⁻ | ≤ 4.14 | 5.88 | 5.67 | 5.04 | 3.44 | 6.59 | 3.28 | 7.27 | 7.16 | 6.63 | 5.61 | 9.39 | 7.02 |
| **ClusComp** | ≤ 4.14 | **5.73** | **5.54** | **4.94** | 3.40 | **6.39** | **3.12** | **7.17** | **7.09** | **6.55** | 5.61 | **9.27** | 6.99 |
| RTN | 3.13 | 7.01 | 6.66 | 5.51 | 3.97 | 27.91 | 11.84 | 8.62 | 8.40 | 7.18 | 6.02 | 27.9 | 22.39 |
| GPTQ | 3.00 | 8.06 | 8.37 | 6.44 | 4.82 | 13.0 | - | 9.49 | 9.81 | 8.02 | 6.57 | 13.00 | - |
| OmniQuant | 3.00 | 6.49 | 6.58 | 5.58 | 3.92 | - | - | 8.19 | 8.65 | 7.44 | 6.06 | - | - |
| AffineQuant | 3.00 | 6.30 | 6.55 | 5.62 | - | - | - | 8.03 | 8.57 | 7.56 | - | - | - |
| QuIP | 3.00 | - | - | - | 3.85 | 7.50 | - | - | - | - | 6.14 | - | - |
| ClusComp⁻ | ≤ 2.89 | 6.74 | 6.54 | 6.27 | 4.02 | 8.77 | 4.98 | 8.14 | 8.19 | 8.21 | 6.06 | 12.41 | 8.26 |
| **ClusComp** | ≤ 2.89 | **6.01** | **5.86** | **5.18** | **3.72** | **7.34** | **4.63** | **7.64** | **7.61** | **6.91** | **5.86** | **11.31** | 8.26 |
| GPTQ | 2.13 | 44.01 | 36.77 | 28.14 | NAN | 2.1e2 | 11.90 | 27.71 | 33.70 | 20.97 | NAN | 2.1e2 | - |
| SliM-LLM⁺ | 2.13 | 9.68 | 10.87 | 7.59 | 6.44 | - | - | 14.99 | 18.18 | 10.24 | 8.40 | - | - |
| QuIP | 2.13 | - | 39.73 | 13.48 | 6.64 | 84.97 | 13.03 | - | 31.94 | 16.16 | 8.17 | 1.3e2 | 22.24 |
| PB-LLM | 2.13 | - | 25.37 | 49.81 | NAN | 44.12 | 11.68 | - | 29.84 | 19.82 | 8.95 | 79.21 | 33.91 |
| GPTVQ | 2.13 | - | 8.23 | 6.50 | 4.64 | - | - | - | - | - | - | - | - |
| AffineQuant | 2.13 | 13.51 | 10.87 | 7.64 | - | - | - | - | 16.02 | 10.98 | - | - | - |
| OmniQuant | 2.14 | 9.72 | 11.06 | 8.26 | 6.55 | - | - | 12.97 | 15.02 | 11.05 | 8.52 | - | - |
| ClusComp⁻ | ≤ 2.15 | 28.76 | 21.90 | 14.50 | 5.43 | 2.1e2 | 11.40 | 29.67 | 25.26 | 18.83 | 7.59 | 1.9e2 | 16.52 |
| **ClusComp** | ≤ 2.15 | **7.06** | **7.04** | **5.85** | **4.37** | **11.57** | **7.61** | **9.33** | **9.49** | **7.92** | **6.44** | **17.89** | **10.81** |
| GPTQ | 2.00 | 2.1e3 | 7.7e3 | 2.1e3 | 77.95 | 5.7e4 | - | 6.9e2 | NAN | 3.2e2 | 48.82 | 5.7e4 | - |
| QuIP | 2.00 | - | - | - | 6.33 | 85.10 | - | - | - | - | - | 1.3e2 | - |
| AffineQuant | 2.00 | 9.53 | 35.07 | 12.42 | - | - | - | - | - | - | - | - | - |
| OmniQuant | 2.00 | 15.47 | 37.37 | 17.21 | 7.81 | - | - | 24.89 | 90.64 | 26.76 | 12.28 | 8.2e5 | - |
| ClusComp⁻ | ≤ 2.01 | 65.09 | 52.38 | 22.90 | 9.84 | 3.1e2 | - | 74.61 | 50.08 | 24.47 | 13.96 | 2.2e2 | - |
| **ClusComp** | ≤ 2.01 | **7.49** | **7.50** | **6.17** | **4.83** | **12.33** | - | **10.11** | **10.29** | **8.49** | **7.02** | **21.45** | - |

evaluation, we apply 6 tasks from lm-eval v0.4.4 (Gao et al., 2024), i.e. PIQA (Bisk et al., 2020), ARC-e/c (Clark et al., 2018), BoolQ (Clark et al., 2019), HellaSwag (Zellers et al., 2019) and Wino-Grande (Sakaguchi et al., 2020). For language modeling, we report the perplexity on the whole test set of WikiText2 (Merity et al., 2017) and on 256 samples from the validation set of C4 (Raffel et al., 2020a) with a sequence length of 2048 as our baselines. We also apply ClusComp to LLaVA-Next-8B (Li et al., 2024b), and evaluate it on 5 multimodal tasks from lmms-eval v0.2.3 (Li et al., 2024a) to show its broad applicability, i.e. AI2D (Kembhavi et al., 2016), ChartQA (Masry et al., 2022), DocVQA (Mathew et al., 2021), MMBench (Liu et al., 2023) and MME (Yin et al., 2023).[3]

**Baselines.** Here we primarily compare ClusComp with three categories of baselines: (1) statistic-based methods without neural training, including vanilla RTN, GPTQ (Frantar et al., 2022), AWQ (Lin et al., 2024), and PB-LLM (Yuan et al., 2024);[4] (2) gradient-based methods with neural training (such as block-wise distillation), including OmniQuant (Shao et al., 2024), AffineQuant (Ma et al., 2024), and SliM-LLM⁺ (Huang et al., 2024b); and (3) quantized codebook-based methods, including QuIP (Chee et al., 2023) and GPTVQ (van Baalen et al., 2024). All baseline results are directly borrowed from the original works or their follow-up works.

**Settings.** We begin by applying K-means clustering to the weight matrices of all linear layers, referring to this method as ClusComp⁻. Next, we use 128 calibration sentences from the Wiki-Text2 training set to minimize block-wise error through codebook training only, which we denote as ClusComp. It is important to highlight that the majority of the aforementioned baselines utilize comparable resources (GPU memory and the number of calibration sentences) to those used in ClusComp. All detailed experimental settings in this section are provided in §B.

**Results.** The language modeling results are presented in Table 2. At the 4-bit level, ClusComp demonstrates superior performance by achieving the lowest perplexity in 9 out of 12 cases, while maintaining a negligible perplexity difference ($\leq 0.05$) compared to the best baselines in the remain-

---

[3]We observed that different studies may report varying zero-shot accuracy for the FP16 model, which can be attributed to the lm-eval version or the choice of evaluation metric (accuracy or normalized accuracy). We recommend that future researchers first reproduce the FP16 accuracy before making comparisons.

[4]ClusComp⁻ is also a statistic-based method.

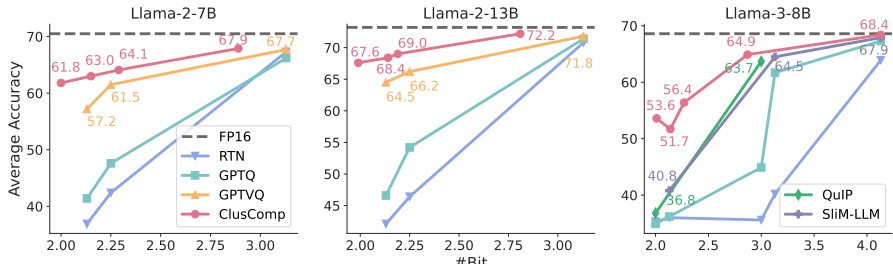

Figure 4: Average zero-shot accuracy over 5/6 commonsense reasoning tasks, only including competitive baselines. Please refer to Table C.3 and C.4 for detailed numbers and the full comparison.

Table 3: Zero-shot multimodal evaluation, with baseline results from Huang et al. (2024c).

| Method | #Bit | AI2D ↑ | ChartQA ↑ | DocVQA ↑ | MMBench ↑ | Avg ↑ | MME (cog / per) ↑ |
|---|---|---|---|---|---|---|---|
| **LLaVA-Next-8B** | 16.00 | 71.7 | 69.2 | 78.2 | 72.2 | 72.8 | 1965.1 (376.8 / 1588.3) |
| GPTQ | 4.13 | **70.7** | 67.4 | 77.4 | 71.0 | 71.6 | 1895.0 (331.6 / 1563.4) |
| AWQ | 4.13 | 70.6 | 68.0 | 77.2 | **71.1** | 71.7 | 1888.4 (325.7 / 1562.7) |
| **ClusComp** | 4.13 | 70.0 | **68.7** | **77.6** | **71.1** | **71.8** | **1915.7** (322.1 / 1593.6) |
| GPTQ | 3.13 | 66.2 | 65.1 | **75.6** | 67.4 | 68.6 | 1831.8 (290.1 / 1541.7) |
| AWQ | 3.13 | 67.7 | 65.4 | 74.4 | **68.0** | 68.9 | 1840.3 (298.6 / 1541.7) |
| **ClusComp** | 2.87 | **68.7** | **65.8** | 74.8 | 67.7 | **69.3** | **1872.6** (331.1 / 1541.5) |
| GPTQ | 2.13 | 0.0 | 0.0 | 0.0 | 0.0 | 0.0 | 0.0 (0.0 / 0.0) |
| AWQ | 2.13 | 0.0 | 0.0 | 0.0 | 0.0 | 0.0 | 0.0 (0.0 / 0.0) |
| **ClusComp** | 2.14 | **53.9** | **53.1** | **56.7** | **50.1** | **53.5** | **1673.0** (294.6 / 1378.4) |

ing 3 cases. At bit-widths $< 4$, ClusComp consistently outperforms all baselines. Notably, even at the 2-bit level, ClusComp's perplexity remains within a functional range, $< 13$ on Wikitext2. Figure 4 presents the zero-shot evaluation results, where ClusComp again consistently surpasses all baselines across different bit-widths. Furthermore, ClusComp exhibits significantly less sensitivity to bit-width variations, as indicated by the flatter slope of its accuracy curve.

We also compress the Llama-3-8B backbone in LLaVA-Next-8B, and report the zero-short performance in Table 3. On average, ClusComp continues to outperform both GPTQ and AWQ, while using a comparable or even lower number of bits. A particularly noteworthy observation occurs at the 2-bit level, where none of the baselines produce correct outputs, whereas ClusComp retains strong performance. In comparison to the 2-bit results for Llama-3-8B in Figure 4 (Right), this suggests that quantizing multimodal models presents unique challenges, warranting further study.

## 4.2 PUSH THE LIMIT OF MODEL COMPRESSION

We further enhance the performance of 2-bit LLMs and extend the compression boundary to the 1-bit level through efficient recovery training. This is achieved by optimizing the codebook parameters in an end-to-end manner during a next-token prediction task.

**Baselines.** We include BiLLM (Huang et al., 2024a), which performs effectively at the 1-bit compression level. Additionally, three more resource-intensive PTQ baselines are considered: AQLM (Egiazarian et al., 2024), which utilizes a larger number of calibration samples (4-16M tokens); QuIP# (Tseng et al., 2024) and DB-LLM (Chen et al., 2024), both of which employ model-wise distillation and a larger number of calibration samples (24-48M tokens).

**Settings.** ClusComp employs only 0.3M tokens for its compression. In this experiment, we further finetune the compressed LLM generated by ClusComp through end-to-end training, optimizing the codebook parameters using 16M tokens from a subset of the RedPajama dataset (Computer, 2023). This extended method is referred to as ClusComp$^+$.

**Results.** We report the zero-shot accuracy of ultra-low-bit LLMs in Table 4. On both Llama-2-7B and Llama-2-13B, ClusComp already performs comparably to, or surpasses AQLM and QuIP#. With minimal recovery training, ClusComp$^+$ consistently outperforms these baselines on aver-

Table 4: Zero-shot accuracy on ultra-low-bit LLMs.

| Method | #Bit | PIQA | ArcE | ArcC | Hella. | Wino. | Avg |
|---|---|---|---|---|---|---|---|
| **Llama-2-7B** | 16.00 | 78.1 | 76.3 | 43.4 | 57.1 | 69.1 | 64.8 |
| QuIP# | 2.02 | **75.1** | 64.6 | **34.6** | 48.3 | 64.9 | 57.5 |
| AQLM | 2.02 | 73.6 | 61.9 | 33.3 | **49.5** | 64.2 | 56.5 |
| ClusComp | 2.00 | 72.6 | 67.0 | 32.9 | 47.2 | 63.4 | 56.6 |
| **ClusComp+** | 2.00 | 73.5 | **67.2** | 33.9 | 49.3 | **65.1** | **57.8** |
| **Llama-2-13B** | 16.00 | 79.1 | 79.4 | 48.4 | 60.0 | 72.2 | 67.8 |
| QuIP# | 2.01 | **77.3** | 69.3 | 39.5 | 53.4 | 67.7 | 61.5 |
| AQLM | 1.97 | 76.2 | 69.8 | 37.8 | 53.7 | 65.4 | 60.6 |
| ClusComp | 1.99 | 75.6 | 74.7 | 39.9 | 53.0 | 67.1 | 62.0 |
| **ClusComp+** | 1.99 | 76.8 | **74.9** | **40.7** | 54.5 | 68.4 | **63.1** |
| **Llama-3-8B** | 16.00 | 79.7 | 80.1 | 50.4 | 60.2 | 72.6 | 68.6 |
| QuIP | 2.00 | 52.9 | 29.0 | 21.3 | 29.2 | 51.7 | 36.8 |
| PB-LLM | 2.00 | 57.0 | 37.8 | 17.2 | 29.8 | 52.5 | 38.8 |
| DB-LLM | 2.00 | 68.9 | 59.1 | 28.2 | 42.1 | 60.4 | 51.8 |
| ClusComp | 2.01 | 70.1 | 63.3 | 31.9 | 44.4 | 58.4 | 53.6 |
| **ClusComp+** | 2.01 | **74.8** | **66.7** | **34.8** | **49.6** | **63.4** | **57.8** |
| **Llama-3-70B** | 16.00 | 82.5 | 86.7 | 60.4 | 66.3 | 80.9 | 75.4 |
| QuIP | 2.00 | 65.3 | 48.9 | 26.5 | 40.9 | **61.7** | 48.7 |
| PB-LLM | 1.70 | 56.5 | 49.9 | 25.8 | 34.9 | 53.1 | 44.1 |
| BiLLM | 1.10 | 58.2 | 46.4 | 25.1 | 37.5 | 53.6 | 44.2 |
| ClusComp | 1.14 | 56.9 | 32.5 | 20.6 | 32.3 | 51.6 | 39.7 |
| **ClusComp+** | 1.14 | **69.5** | **57.2** | **30.1** | **44.2** | 56.0 | **51.4** |

Table 5: In-domain finetuning performance on Llama-2-7B. Two bits are shown for baselines, since the LoRA modules aren't merged to the quantized LLMs. The first and second numbers denote the quantized LLM and the converted bits from LoRA modules.

| Method | #Bit | WikiText2 PPL ↓ | GSM8K ACC ↑ |
|---|---|---|---|
| LoRA | 16.00 | 5.08 | 36.9 |
| QLoRA | 4.25 + 0.40 | 5.70 | 35.1 |
| LoftQ | 4.25 + 0.40 | **5.24** | 35.0 |
| **ClusComp** | 4.15 | 5.26 | **41.0** |
| QLoRA | 3.25 + 0.40 | 5.73 | 32.1 |
| LoftQ | 3.25 + 0.40 | 5.63 | 32.9 |
| **ClusComp** | 3.38 | **5.37** | **39.9** |
| QLoRA | 2.25 + 0.40 | NAN | NAN |
| LoftQ | 2.25 + 0.40 | 7.85 | 20.9 |
| **ClusComp** | 2.54 | **5.78** | **37.2** |
| **ClusComp** | 2.29 | 6.10 | 36.0 |

Table 6: General-domain finetuning performance, with baseline results from Xu et al. (2024b).

| Method | #Bit | MMLU (0-shot, ACC ↑) | | | | | MMLU (5-shot, ACC ↑) | | | | |
|---|---|---|---|---|---|---|---|---|---|---|---|
| | | Hums. | STEM | Social | Other | Avg | Hums. | STEM | Social | Other | Avg |
| **Llama-1-7B** | 16.00 | 32.4 | 26.6 | 31.4 | 37.2 | 32.1 | 33.3 | 29.8 | 37.8 | 38.0 | 34.6 |
| GPTQ-LoRA | 4.50 | 35.7 | 30.9 | 38.0 | 44.0 | 37.1 | 33.8 | 31.3 | 37.4 | 42.2 | 36.0 |
| QA-LoRA | 4.50 | **36.9** | **31.4** | **40.3** | **44.9** | **38.3** | 36.6 | 32.4 | **44.8** | **44.9** | **39.4** |
| PEQA | 4.00 | - | - | - | - | - | 34.9 | 28.9 | 37.5 | 40.1 | 34.8 |
| **ClusComp** | 4.15 | **36.9** | **31.4** | 40.0 | 44.2 | 38.0 | **36.8** | **34.1** | 42.7 | 43.9 | 39.1 |
| GPTQ-LoRA | 3.50 | 31.5 | 28.9 | 31.8 | 36.8 | 32.2 | 31.6 | 30.1 | 35.6 | 39.8 | 34.0 |
| QA-LoRA | 3.50 | 36.0 | **34.1** | **42.0** | 42.3 | 38.3 | 35.6 | 30.5 | **41.5** | 42.7 | 37.4 |
| **ClusComp** | 3.38 | **38.2** | 32.7 | 41.2 | **45.4** | **39.2** | **36.3** | **31.4** | 41.3 | **43.0** | **37.8** |
| GPTQ-LoRA | 2.50 | 24.1 | 22.1 | 22.5 | 23.7 | 23.2 | 23.4 | 26.2 | 26.4 | 28.4 | 25.8 |
| QA-LoRA | 2.50 | 26.4 | 25.5 | 25.6 | 28.7 | 26.5 | 27.3 | 26.1 | 26.1 | 30.3 | 27.5 |
| **ClusComp** | 2.29 | **32.6** | **29.7** | **34.4** | **37.0** | **33.3** | **31.1** | **30.1** | **37.8** | **37.2** | **33.7** |
| **Llama-2-7B** | 16.00 | 38.9 | 32.9 | 46.6 | 44.9 | 40.7 | 43.0 | 36.4 | 51.4 | 52.2 | 45.5 |
| QA-LoRA | 4.50 | 41.1 | 35.4 | 50.2 | 50.1 | 43.9 | 42.1 | 34.4 | 49.1 | 50.3 | 43.9 |
| **ClusComp** | 4.15 | **41.6** | **36.3** | **52.3** | **51.1** | **44.9** | **42.8** | **38.1** | **52.2** | **53.1** | **46.1** |
| **Llama-2-13B** | 16.00 | 48.1 | 42.7 | 60.5 | 59.5 | 52.3 | 53.3 | 44.1 | 63.3 | 61.0 | 55.3 |
| QA-LoRA | 4.50 | 48.2 | 41.7 | 60.4 | 58.7 | 51.9 | 48.0 | 43.0 | 59.7 | 57.4 | 51.7 |
| **ClusComp** | 4.09 | **49.2** | **42.9** | **61.6** | **60.2** | **52.9** | **52.4** | **43.2** | **62.9** | **61.6** | **54.7** |

age, with the performance gap increasing for larger LLMs, indicating the robust scalability of ClusComp+. On Llama-3-8B, ClusComp already exceeds all baselines, and ClusComp+ further widens this margin. On Llama-3-70B, ClusComp+ achieves remarkable accuracy at the 1-bit level. Furthermore, when comparing the improvements from ClusComp to ClusComp+ across different Llama series, a notably larger performance gain is observed on the Llama-3 models, underscoring the effectiveness of ClusComp+ on LLMs with a higher frequency of outliers.

## 4.3 FINETUNING QUALITY AND EFFICIENCY

We can finetune the compressed LLMs on downstream tasks by only training the codebooks.

**In-domain finetuning.** We finetune Llama-2-7B on the training sets of WikiText2 and GSM8K (Cobbe et al., 2021), and report the perplexity and accuracy on their respective validation/test set. ClusComp is compared with two LoRA-based techniques: QLoRA (Dettmers et al., 2023) and LoftQ (Li et al., 2024c). As shown in Table 5, ClusComp consistently achieves superior results with fewer bits (except at the 4-bit level on WikiText2, where it performs comparably to LoftQ), even outperforming LoRA-finetuning of the FP16 model on GSM8K with a 2.54-bit compressed LLM.

Table 7: ClusComp performance against efficient full finetuning, with baseline results from (Pan et al., 2024).

| Method | Bit | #Trained | MMLU 5-shot ↑ | AGIEval 3-shot ↑ |
|---|---|---|---|---|
| **Llama-2-7B** | 16.00 | - | 45.9 | 25.7 |
| Full FT | 16.00 | 100% | 45.7 | 27.0 |
| LoRA ($r = 128$) | 16.00 | 4.9% | 45.5 | 24.7 |
| GaLore | 16.00 | 100.0% | 45.5 | 24.4 |
| LISA | 16.00 | 100.0% | 46.2 | 26.1 |
| **ClusComp** | 4.15 | 0.9% | **47.0** | **26.5** |
| **ClusComp** | 2.88 | 1.4% | 45.1 | 25.6 |
| **ClusComp** | 2.00 | 1.4% | 30.7 | 21.8 |

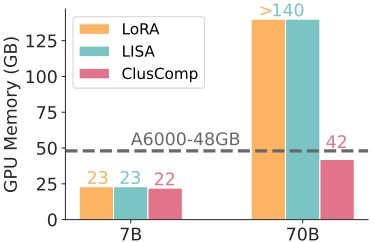

Figure 5: Memory consumption for recovery training or finetuning. 4-bit LLMs are used for ClusComp here.

**General-domain finetuning.** We finetune the compressed LLMs on Alpaca-GPT3.5 (Taori et al., 2023) and evaluate them using the MMLU benchmark (Hendrycks et al., 2021). ClusComp is compared against baseline methods that merge trained LoRA modules into the quantized linear layers after finetuning, i.e. GPTQ-LoRA, QA-LoRA (Xu et al., 2024b) and PEQA (Kim et al., 2023).

As shown in Table 6, ClusComp consistently outperforms the baseline methods across different LLMs and bit-widths, while using fewer bits. The only exception occurs at the 4-bit level for Llama-1-7B, where ClusComp underperforms QA-LoRA by a small margin of 0.3 accuracy. The performance gap between ClusComp and baselines is enlarged for lower bits or recent LLM series.

**Compared to full finetuning.** Similar to the general-domain finetuning, we finetune the compressed LLMs on a new version of Alpaca, i.e. Alpaca-GPT4 (Peng et al., 2023), and evaluate them on both MMLU and AGIEval (Zhong et al., 2024). Here we mainly compare ClusComp to some memory-efficient finetuning methods that fully finetune the FP16 version, i.e. GaLore (Zhao et al., 2024) and LISA (Pan et al., 2024). As shown in Table 7, finetuning the compressed LLMs at the 4-bit level from ClusComp outperforms all memory-efficient finetuning methods, and rivals full finetuning. In addition, the finetuned LLMs can be used in a low bit, friendly for inference.

The superior finetuning performance can be attributed to three key advantages of ClusComp: (1) ClusComp introduces smaller compression errors; (2) Unlike QLoRA, where compressed LLMs are frozen during finetuning, ClusComp allows for the model to remain unfrozen by training the codebook parameters, enabling further mitigation of compression errors; (3) The low-rank design of LoRA limits its expressiveness. In contrast, updating the codebook in ClusComp is analogous to updating a high-rank weight matrix. In addition, the fixed-code design allows uniform training of all centroids, providing greater expressiveness and can even rival full finetuning.

**Efficiency discussion.** Figure 5 illustrates the memory efficiency of ClusComp during training, with a batch size of 1 and a sequence length of 1024. For the 70B LLM, we apply gradient checkpointing (this is not used for the 7B LLM), while omitting any additional memory-saving techniques.

Finetuning the LLM compressed by ClusComp demonstrates memory efficiency in two key ways: (1) The compressed LLM requires less memory for loading onto the GPU compared to the FP16 model; and (2) Only the codebook parameters, which contain a limited number of trainable parameters ($< 1\%$), are updated, as detailed in Table 7. Consequently, the optimizer state size remains small. ClusComp can serve not only as a model compression technique but also as an effective method for both memory- and parameter-efficient finetuning.

## 5 CONCLUSION

The newly introduced model compression technique, ClusComp, operates by (1) independently applying clustering to the weight matrices to produce both the codebook and corresponding codes, (2) reducing compression error through block-wise knowledge distillation, and (3) enhancing model performance via efficient recovery finetuning. Comprehensive experiments demonstrate its effectiveness as a compression method at 1-4 bit levels, while also showcasing its parameter and memory efficiency for finetuning, with a competitive performance with full finetuning.

REPRODUCIBILITY STATEMENT

We explain all experimental details in Section §B, and guarantee the open source of our code upon decision notification.

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

## A LIMITATION AND FUTURE WORK

**Limitation.** ~~Although ClusComp demonstrates strong performance in both model compression and finetuning tasks, its inference speed remains similar to that of FP16 models. As illustrated in Listing C.1, ClusComp introduces two additional operations—indexing and reshaping—beyond those found in a standard linear layer. These operations are computationally efficient, resulting in an inference speed that is similar to that of the original linear layer. Since no quantization techniques are applied, the transfer of weight tensors does not contribute to time savings, resulting in a smaller inference speed than the uniform quantization methods. Nevertheless, we consider this trade-off acceptable given the model's notable performance in compression and finetuning.~~

**Future work.** The following list of tasks is in our plan:

- ~~Design a new CUDA kernel that is more efficient for ClusComp.~~
- Apply ClusComp to reduce the memory requirement for caching the keys and values, which facilitates LLMs for long-context tasks.
- Apply the fixed-code idea to a VAE-like method to scale up the size and dimension of the codebook for the image generation task.
- Study the quantization of large multimodal models, since they show different behaviors from LLMs (see Table 3).

## B EXPERIMENTAL DETAILS

### B.1 CLUSTERING

We use the K-means implementation from the Faiss library (Douze et al., 2024). The number of iterations is set to 20, with all default settings for other arguments.

### B.2 BLOCK-WISE ERROR MINIMIZATION

For all LLMs, 128 calibration sentences with a length of 2048 tokens are randomly selected from the WikiText-2 training set (Merity et al., 2017). The detailed hyper-parameters are listed in Table B.1. Only the codebooks are trained, while keeping all other parameters (from the embedding layer, output layer and normalization layers) frozen.

Table B.1: Hyper-parameters used for the block-wise error minimization and recovery training steps. The underlined settings generally perform well for different scales of LLMs.

| Hyper-parameter | Block-wise error minimization | Recovery training |
|---|---|---|
| Optimizer | AdamW (Loshchilov & Hutter, 2019; Kingma & Ba, 2015) | |
| Weight decay | {0, 0.1, 0.01} | 0 |
| LR | {1e-5, 5e-5, 1e-4, 5e-4} | 1e-5 |
| LR scheduler | constant | cosine |
| Warmup ratio | 0 | 0 |
| Max grad norm | - | 0.3 |
| Sequence length | 2048 | 4096 |
| Number of samples | 128 | 8192 |
| Epochs | 20 | 1 |
| Batch size | 8 | 8 |

### B.3 RECOVERY TRAINING

For the recovery training, we randomly sample 1024 sentences with a length of 4096 tokens from a subset of RedPajama (Computer, 2023).[5] Then we train the compressed LLMs to predict the next token by only tuning the codebook parameters. The hyperparameters used in this step are listed in Table B.1.

---

[5]https://huggingface.co/datasets/togethercomputer/RedPajama-Data-1T-Sample

## B.4 IN-DOMAIN FINETUNING

We follow the settings from (Li et al., 2024c), and finetune the compressed LLM on the training set of WikiText2 and on the training set of GSM8K. The hyperparameters for finetuning are listed in Table B.2. We evaluate the finetuned model on the validation set of WikiText2 and on the test set of GSM8K every epoch and report the best perplexity or accuracy.

Table B.2: Hyperparameters for the finetuning on Llama-2-7B. The underlined settings generally performs well for different bit levels.

| Hyper-parameter | WikiText-2 | GSM8K | Alpaca-GPT3.5 | Alpaca-GPT4 |
|---|---|---|---|---|
| Optimizer | AdamW | | AdamW | |
| Weight decay | 0.1 | | 0 | |
| LR | $\{0.7, \underline{1}, 3\} \times 10^{-4}$ | | $\{2, 4, \underline{6}, 8\} \times 10^{-5}$ | |
| LR scheduler | cosine | | cosine | |
| Warmup ratio | 3% | | 6% | |
| Epochs or max steps | 3 epochs | 6 epochs | 10K steps | 2 epochs |
| Batch size | 64 | 16 | 16 | |
| Max sequence length | 1024 | 512 | 2048 | |

## B.5 GENERAL-DOMAIN FINETUNING

The finetuning hyper-parameters are listed in Table B.2, which is similar to the ones in QA-LoRA (Xu et al., 2024b) on Alpaca-GPT3.5, or to the ones in LISA (Pan et al., 2024) on Alpaca-GPT4.

## C MORE RESULTS

In this section, we provide the detailed numbers for the figures in the main pages and more results:

- We present the kurtosis of various types of layers in Figure C.1, as a complement to Figure 2 (Right). We hypothesize that the higher kurtosis observed in Llama-3 may be attributed to two factors: the larger pretraining steps (Bondarenko et al., 2023) and the inclusion of multilingual data. However, as this is beyond the scope of the current study, we defer further investigation to future work.
- The modified linear layer is illustrated in Listing C.1.
- We present the full perplexity results on WikiText2 and C4 in Table C.1 and C.2, as a complement to Table 2.
- We present the full zero-shot evaluation accuracy and the reported metrics in Table C.3 and C.4, as a complement to Figure 4.
- The quantization quality of the Llama-3-8B backbone in LLaVA-Next-8B on WikiText2 and C4 is shown in Table C.5.

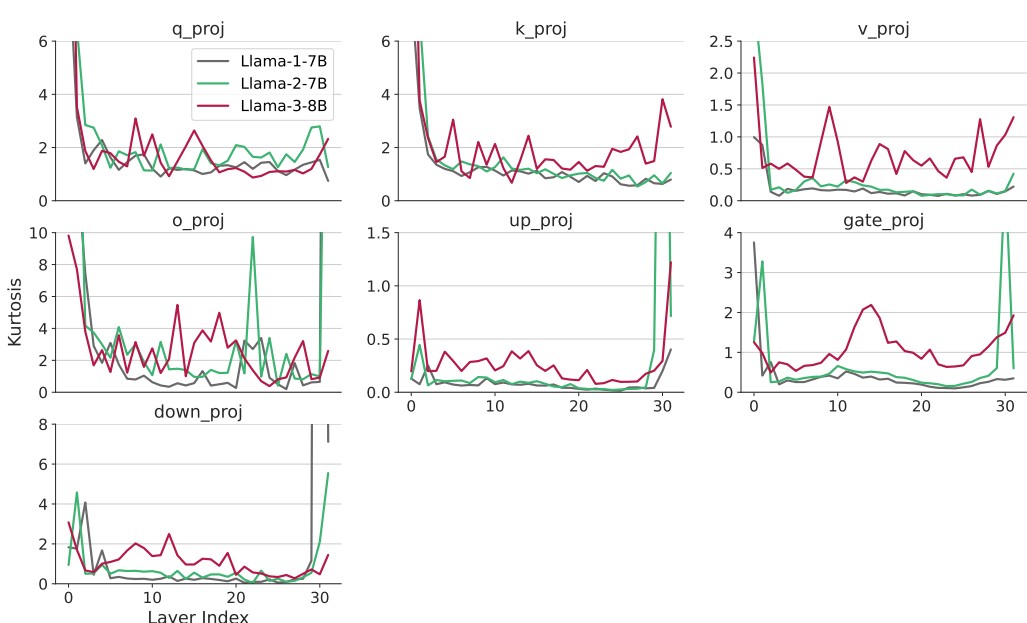

Figure C.1: The kurtosis across various layers in different Llama series reveals three key observations: (1) Layers at either the beginning or the end of LLMs tend to exhibit higher kurtosis values; (2) In the majority of layers, the kurtosis follows a consistent trend across Llama series, with Llama-3 showing the highest values, followed by Llama-2, and then Llama-1; (3) Different types of layers display varying scales of kurtosis, suggesting that a bit allocation strategy that accounts for quantization difficulty could yield better results. We leave the exploration of this idea to future work.

```python
class ClusCompLinear(nn.Module):
    def __init__(self, in_features, out_features, num_clusters, cluster_dim, bias):
        super().__init__()
        self.out_features = out_features
        self.in_features = in_features
        self.deficiency = out_features % cluster_dim # If the out_features is not dividable by cluster_dim
        if self.deficiency > 0:
            self.deficiency = cluster_dim - self.deficiency

        num_codes = in_features * (out_features + self.deficiency) // cluster_dim
        self.codebook = nn.Parameter(torch.empty((num_clusters, cluster_dim), dtype=torch.bfloat16) #
            trainable
        code = torch.empty((num_codes,), dtype=torch.uint16)
        self.register_buffer('code', code) # non-trainable
        if bias:
            self.bias = nn.Parameter(torch.empty(out_features))
        else:
            self.register_parameter('bias', None)

    def forward(self, x):
        vectors = self.codebook[self.code]
        if self.deficiency > 0:
            weight = vectors.view(self.in_features, -1)[:, :-self.deficiency]
        else:
            weight = vectors.view(self.in_features, -1)

        if self.bias is not None:
            out = torch.matmul(x, weight) + self.bias
        else:
            out = torch.matmul(x, weight)
        return out
```

Listing C.1: PyTorch code for the linear layer of ClusComp. All data type is 16-bit.

Table C.1: The full perplexity results of Llama series on WikiText2. "g" and "n" denote the dimension and number of centroids in the codebook, respectively. The number in the brackets is the exact bits of different settings for different LLMs.

| Method | Setting | #Bit | 1-7B | 2-7B | 2-13B | 2-70B | 3-8B | 3-70B |
|---|---|---|---|---|---|---|---|---|
| - | - | 16.00 | 5.68 | 5.47 | 4.88 | 3.31 | 6.12 | 2.90 |
| RTN | w4g128 | 4.13 | 5.96 | 5.72 | 4.98 | 3.46 | 8.50 | 3.60 |
| GPTQ | w4g128 | 4.13 | 5.85 | 5.61 | 4.98 | 3.42 | 6.50 | 3.30 |
| AWQ | w4g128 | 4.13 | 5.81 | 5.62 | 4.97 | - | 6.60 | 3.30 |
| GPTVQ | w4g128 | 4.13 | - | 5.68 | 4.97 | **3.39** | - | - |
| AffineQuant | w4g128 | 4.13 | 5.77 | 5.58 | 4.95 | - | - | - |
| OmniQuant | w4g128 | 4.16 | 5.77 | 5.58 | 4.95 | 3.40 | - | - |
| ClusComp⁻ | g4n65500 | ≤4.14 | 5.88 (4.14) | 5.67 (4.14) | 5.04 (4.09) | 3.44 (4.03) | 6.59 (4.13) | 3.28 (4.03) |
| **ClusComp** | g4n65500 | ≤4.14 | **5.73** (4.14) | **5.54** (4.14) | **4.94** (4.09) | **3.40** (4.03) | **6.39** (4.13) | **3.12** (4.03) |
| RTN | w4 | 4.00 | 6.43 | 6.11 | 5.20 | 3.67 | 8.70 | - |
| GPTQ | w4 | 4.00 | 6.13 | 5.83 | 5.13 | 3.58 | **7.00** | - |
| AWQ | w4 | 4.00 | 6.08 | 6.15 | 5.12 | - | 7.10 | - |
| QuIP | w4 | 4.00 | - | - | - | 3.53 | - | - |
| AffineQuant | w4 | 4.00 | **5.84** | 5.69 | **5.01** | - | - | - |
| OmniQuant | w4 | 4.00 | 5.86 | 5.74 | 5.02 | **3.47** | - | - |
| ClusComp⁻ | g5n65500 | 3.38 | 6.27 | 5.90 | - | - | - | - |
| **ClusComp** | g5n65500 | 3.38 | **5.84** | **5.67** | - | - | - | - |
| RTN | w3g128 | 3.13 | 7.01 | 6.66 | 5.51 | 3.97 | 27.91 | 11.84 |
| GPTQ | w3g128 | 3.13 | 6.55 | 6.29 | 5.42 | 3.85 | 8.22 | 5.22 |
| AWQ | w3g128 | 3.13 | 6.46 | 6.24 | 5.32 | - | **8.19** | **4.81** |
| SliM-LLM⁺ | w3g128 | 3.13 | **6.07** | 5.94 | 5.11 | **3.35** | - | - |
| AffineQuant | w3g128 | 3.13 | 6.14 | 6.08 | 5.28 | - | - | - |
| GPTVQ | w3g128 | 3.13 | - | **5.82** | **5.10** | 3.55 | - | - |
| OmniQuant | w3g128 | 3.15 | 6.15 | 6.03 | 5.28 | 3.78 | - | - |
| RTN | w3 | 3.00 | 25.73 | 5.4e2 | 10.68 | 7.52 | 2.2e3 | - |
| GPTQ | w3 | 3.00 | 8.06 | 8.37 | 6.44 | 4.82 | 13.0 | - |
| AWQ | w3 | 3.00 | 11.88 | 24.00 | 10.45 | - | 12.8 | - |
| QuIP | w3 | 3.00 | - | - | - | 3.85 | 7.5 | - |
| AffineQuant | w3 | 3.00 | 6.30 | 6.55 | 5.62 | - | - | - |
| OmniQuant | w3 | 3.00 | 6.49 | 6.58 | 5.58 | 3.92 | - | - |
| ClusComp⁻ | g6n65500 | ≤2.89 | 6.74 (2.89) | 6.54 (2.89) | 6.27 (2.81) | 4.02 (2.72) | 8.77 (2.87) | 4.98 (2.72) |
| **ClusComp** | g6n65500 | ≤2.89 | **6.01** (2.89) | **5.86** (2.89) | **5.18** (2.81) | **3.72** (2.72) | **7.34** (2.87) | **4.63** (2.72) |
| ClusComp⁻ | g7n65500 | 2.54 | 7.79 | 7.64 | - | - | - | - |
| **ClusComp** | g7n65500 | 2.54 | **6.28** | **6.15** | - | - | - | - |
| RTN | w2g64 | 2.25 | 1.9e2 | 4.3e2 | 26.22 | 10.31 | - | - |
| GPTQ | w2g64 | 2.25 | 22.10 | 20.85 | 22.44 | NAN | 1.8e2 | - |
| AWQ | w2g64 | 2.25 | 2.5e5 | 2.1e5 | 1.2e5 | - | - | - |
| GPTVQ | w2g64 | 2.25 | - | 7.22 | 6.08 | **4.39** | - | - |
| AffineQuant | w2g64 | 2.25 | 8.35 | 9.05 | 7.11 | - | - | - |
| OmniQuant | w2g64 | 2.28 | 8.90 | 9.62 | 7.56 | 6.11 | - | - |
| ClusComp⁻ | g8n65500 | ≤2.29 | 10.25 (2.29) | 11.10 (2.29) | 14.39 (2.19) | - | 29.20 (2.27) | - |
| **ClusComp** | g8n65500 | ≤2.29 | **6.66** (2.29) | **6.61** (2.29) | **5.74** (2.19) | - | **9.68** (2.27) | - |
| RTN | w2g128 | 2.13 | 1.9e3 | 4.2e3 | 1.2e2 | 27.27 | 1.9e3 | 4.6e5 |
| GPTQ | w2g128 | 2.13 | 44.01 | 36.77 | 28.14 | NAN | 2.1e2 | 11.9 |
| AWQ | w2g128 | 2.13 | 2.6e5 | 2.2e5 | 1.2e5 | - | 1.7e6 | 1.7e6 |
| SliM-LLM⁺ | w2g128 | 2.13 | 9.68 | 10.87 | 7.59 | 6.44 | - | - |
| QuIP | w2g128 | 2.13 | - | 39.73 | 13.48 | 6.64 | 84.97 | 13.03 |
| PB-LLM | w2g128 | 2.13 | - | 25.37 | 49.81 | NAN | 44.12 | 11.68 |
| GPTVQ | w2g128 | 2.13 | - | 8.23 | 6.50 | 4.64 | - | - |
| AffineQuant | w2g128 | 2.13 | 13.51 | 10.87 | 7.64 | - | - | - |
| OmniQuant | w2g128 | 2.14 | 9.72 | 11.06 | 8.26 | 6.55 | - | - |
| ClusComp⁻ | g8 | ≤2.15 | 28.76 | 21.9 | 14.50 | 5.43 | 2.1e2 | 11.40 |
| **ClusComp** | g8 | ≤2.15 | **7.06** (2.15,n35000) | **7.04** (2.15,n35000) | **5.85** (2.14,n50000) | **4.37** (2.07,n65500) | **11.57** (2.14,n35000) | **7.61** (2.07,n65500) |
| ClusComp⁻ | g9n65500 | 2.11 | - | 22.71 | - | - | - | - |
| **ClusComp** | g9n65500 | 2.11 | - | **7.12** | - | - | - | - |
| RTN | w2 | 2.00 | 1.1e5 | 3.8e4 | 5.6e4 | 2.0e4 | 2.7e6 | - |
| GPTQ | w2 | 2.00 | 2.1e3 | 7.7e3 | 2.1e3 | 77.95 | 5.7e4 | - |
| QuIP | w2 | 2.00 | - | - | - | 6.33 | 85.1 | - |
| AffineQuant | w2 | 2.00 | 9.53 | 35.07 | 12.42 | - | - | - |
| OmniQuant | w2 | 2.00 | 15.47 | 37.37 | 17.21 | 7.81 | - | - |
| ClusComp⁻ | g9 | ≤2.01 | 65.09 | 52.38 | 22.90 | 9.84 | 3.1e2 | - |
| **ClusComp** | g9 | ≤2.01 | **7.49** (2.00,n45000) | **7.50** (2.00,n45000) | **6.17** (1.99,n65500) | **4.83** (1.85,n65500) | **12.33** (2.01,n50000) | - |

Table C.2: The full perplexity results of Llama series on C4. "g" and "n" denote the dimension and number of centroids in the codebook, respectively. The number in the brackets is the exact bits of different settings for different LLMs.

| Method | Setting | #Bit | 1-7B | 2-7B | 2-13B | 2-70B | 3-8B | 3-70B |
|---|---|---|---|---|---|---|---|---|
| - | - | 16.00 | 7.08 | 6.97 | 6.46 | 5.52 | 9.20 | 5.87 |
| RTN | w4g128 | 4.13 | 7.37 | 7.24 | 6.58 | 5.63 | 13.40 | 8.90 |
| GPTQ | w4g128 | 4.13 | 7.21 | 7.12 | 6.56 | **5.58** | 10.40 | **6.94** |
| AWQ | w4g128 | 4.13 | 7.21 | 7.13 | 6.56 | - | 9.40 | 7.00 |
| AffineQuant | w4g128 | 4.13 | 7.20 | 7.12 | 6.56 | - | - | - |
| OmniQuant | w4g128 | 4.16 | 7.21 | 7.12 | 6.56 | 5.58 | - | - |
| ClusComp⁻ | g4n65500 | ≤4.14 | 7.27 (4.14) | 7.16 (4.14) | 6.63 (4.09) | 5.61 (4.03) | 9.39 (4.13) | 7.02 (4.03) |
| **ClusComp** | g4n65500 | ≤4.13 | **7.17** (4.14) | **7.09** (4.14) | **6.55** (4.09) | 5.61 (4.03) | **9.27** (4.13) | 6.99 (4.03) |
| RTN | w3 | 3.00 | 28.26 | 4.0e2 | 12.51 | 10.02 | 2.2e3 | - |
| GPTQ | w3 | 3.00 | 9.49 | 9.81 | 8.02 | 6.57 | 13.0 | - |
| AWQ | w3 | 3.00 | 13.26 | 23.85 | 13.07 | - | 12.8 | - |
| QuIP | w3 | 3.00 | - | - | - | 6.14 | - | - |
| AffineQuant | w3 | 3.00 | 8.03 | 8.57 | 7.56 | - | - | - |
| OmniQuant | w3 | 3.00 | 8.19 | 8.65 | 7.44 | 6.06 | - | - |
| ClusComp⁻ | g6n65500 | ≤2.89 | 8.14 (2.89) | 8.19 (2.89) | 8.21 (2.81) | 6.06 (2.72) | 12.41 (2.87) | 8.26 (2.72) |
| **ClusComp** | g6n65500 | ≤2.89 | **7.64** (2.89) | **7.61** (2.89) | **6.91** (2.81) | **5.86** (2.72) | **11.31** (2.87) | **8.26** (2.72) |
| ClusComp⁻ | g7n65500 | 2.54 | 9.46 | 9.51 | - | - | - | - |
| **ClusComp** | g7n65500 | 2.54 | **8.10** | **8.13** | - | - | - | - |
| RTN | w2g64 | 2.25 | 1.5e2 | 4.8e2 | 28.69 | 13.43 | - | - |
| GPTQ | w2g64 | 2.25 | 17.71 | 19.40 | 12.48 | NAN | - | - |
| AWQ | w2g64 | 2.25 | 2.8e5 | 1.6e5 | 9.5e4 | - | - | - |
| OmniQuant | w2g64 | 2.28 | 11.78 | 12.72 | 10.05 | 7.88 | - | - |
| ClusComp⁻ | g8n65500 | ≤2.29 | 13.06 (2.29) | 14.07 (2.29) | 19.75 (2.19) | - | 38.68 (2.27) | - |
| **ClusComp** | g8n65500 | ≤2.29 | **8.76** (2.29) | **8.88** (2.29) | **7.75** (2.19) | - | **15.57** (2.27) | - |
| RTN | w2g128 | 2.13 | 1.0e3 | 4.9e3 | 1.4e2 | 42.13 | 1.9e3 | - |
| GPTQ | w2g128 | 2.13 | 27.71 | 33.70 | 20.97 | NAN | 2.1e2 | - |
| AWQ | w2g128 | 2.13 | 1.9e5 | 1.7e5 | 9.4e4 | - | 1.7e6 | - |
| SliM-LLM+ | w2g128 | 2.13 | 14.99 | 18.18 | 10.24 | 8.40 | - | - |
| QuIP | w2g128 | 2.13 | - | 31.94 | 16.16 | 8.17 | 1.3e2 | 22.24 |
| PB-LLM | w2g128 | 2.13 | - | 29.84 | 19.82 | 8.95 | 79.21 | 33.91 |
| AffineQuant | w2g128 | 2.13 | - | 16.02 | 10.98 | - | - | - |
| OmniQuant | w2g128 | 2.14 | 12.97 | 15.02 | 11.05 | 8.52 | - | - |
| ClusComp⁻ | g8 | ≤2.15 | 29.67 | 25.26 | 18.83 | 7.59 | 1.9e2 | 16.52 |
| **ClusComp** | g8 | ≤2.15 | **9.33** (2.15,n35000) | **9.49** (2.15,n35000) | **7.92** (2.14,n50000) | **6.44** (2.07,n65500) | **17.89** (2.14,n35000) | **10.81** (2.07,n65500) |
| ClusComp⁻ | g9n65500 | 2.11 | - | 27.37 | - | - | - | - |
| **ClusComp** | g9n65500 | 2.11 | - | **9.73** | - | - | - | - |
| RTN | w2 | 2.00 | 1.3e5 | 4.8e4 | 7.2e4 | 2.4e4 | 2.7e6 | - |
| GPTQ | w2 | 2.00 | 6.9e2 | NAN | 3.2e2 | 48.82 | 5.7e4 | - |
| QuIP | w2 | 2.00 | - | - | - | - | 1.3e2 | - |
| OmniQuant | w2 | 2.00 | 24.89 | 90.64 | 26.76 | 12.28 | 8.2e5 | |
| ClusComp⁻ | g9 | ≤2.01 | 74.61 | 50.08 | 24.47 | 13.96 | 2.2e2 | - |
| **ClusComp** | g9 | ≤2.01 | **10.11** (2.00,n45000) | **10.29** (2.00,n45000) | **8.49** (1.99,n65500) | **7.02** (1.85,n65500) | **21.45** (2.01,n50000) | - |

Table C.3: Zero-shot evaluation of the quantized Llama-2-7B and Llama-2-13B, with baseline results taken from van Baalen et al. (2024). "acc" and "acc_n" mean accuracy and normalized accuracy, respectively. We offer the results of all metrics for a convenient comparison of the follow-up works. But only the highlighted **metrics** are used to calculate the average accuracy.

| Method | #Bit | PIQA | | ARC-e | | ARC-c | | BoolQ | HellaSwag | | WinoGrande | Avg |
|---|---|---|---|---|---|---|---|---|---|---|---|---|
| | | acc | acc_n | acc | acc_n | acc | acc_n | acc | acc | acc_n | acc | |
| **Llama-2-7B** | 16.00 | - | 79.1 | - | 74.6 | - | 46.3 | 77.7 | - | 76.0 | 69.1 | 70.5 |
| ClusComp | 4.14 | 77.5 | 79.2 | 75.3 | 72.8 | 42.7 | 45.3 | 76.2 | 56.5 | 75.1 | 68.9 | 69.6 |
| RTN | 3.13 | - | 76.8 | - | 70.5 | - | 42.9 | 71.7 | - | **74.0** | 67.6 | 67.3 |
| GPTQ | 3.13 | - | 77.4 | - | 68.1 | - | 40.7 | 71.0 | - | 72.5 | 67.3 | 66.2 |
| GPTVQ | 3.13 | - | **77.6** | - | **72.7** | - | **43.7** | 71.7 | - | 72.7 | 67.6 | 67.7 |
| **ClusComp** | **2.89** | 76.8 | **77.6** | 74.4 | 71.3 | 42.3 | 42.9 | **74.6** | 54.4 | 72.4 | **68.8** | **67.9** |
| RTN | 2.25 | - | 58.8 | - | 36.7 | - | 24.8 | 41.9 | - | 40.4 | 51.9 | 42.4 |
| GPTQ | 2.25 | - | 60.8 | - | 39.0 | - | 25.2 | 59.3 | - | 45.8 | 55.5 | 47.6 |
| GPTVQ | 2.25 | - | 73.3 | - | 63.4 | - | 35.9 | 66.3 | - | 63.9 | **66.1** | 61.5 |
| **ClusComp** | 2.29 | 74.9 | **76.0** | 69.8 | **65.2** | 37.7 | **37.4** | **73.0** | 51.1 | **68.4** | 65.0 | **64.1** |
| ClusComp | 2.15 | 74.3 | 75.1 | 69.6 | 65.2 | 35.7 | 38.4 | 69.5 | 49.2 | 66.4 | 63.5 | 63.0 |
| RTN | 2.13 | - | 51.1 | - | 28.0 | - | 25.0 | 41.1 | - | 26.6 | 49.9 | 36.9 |
| GPTQ | 2.13 | - | 54.8 | - | 30.6 | - | 25.1 | 53.4 | - | 33.1 | 51.5 | 41.4 |
| GPTVQ | 2.13 | - | 70.7 | - | 58.1 | - | 31.5 | 63.7 | - | 58.5 | 60.9 | 57.2 |
| **ClusComp** | **2.00** | 72.6 | **73.7** | 67.0 | **62.8** | 32.9 | **36.6** | **70.9** | 47.2 | **63.5** | 63.4 | **61.8** |
| **Llama-2-13B** | 16.00 | - | 80.5 | - | 77.5 | - | 49.2 | 80.5 | - | 79.4 | 72.1 | 73.2 |
| ClusComp | 4.09 | 78.9 | 79.9 | 78.9 | 76.9 | 47.7 | 49.2 | 81.4 | 60.0 | 79.0 | 72.4 | 73.1 |
| RTN | 3.13 | - | 78.9 | - | 74.3 | - | 46.8 | 77.3 | - | 76.5 | 70.8 | 70.8 |
| GPTQ | 3.13 | - | 79.3 | - | 75.8 | - | 47.0 | 78.9 | - | **77.2** | 70.4 | 71.4 |
| GPTVQ | 3.13 | - | 79.4 | - | 75.3 | - | **48.1** | 79.0 | - | 77.0 | **71.7** | 71.8 |
| **ClusComp** | **2.81** | 78.7 | **79.7** | 78.5 | **76.7** | 45.9 | 47.8 | **80.7** | 58.3 | 76.8 | 71.4 | **72.2** |
| RTN | 2.25 | - | 61.6 | - | 41.6 | - | 25.4 | 49.8 | - | 48.2 | 51.9 | 46.4 |
| GPTQ | 2.25 | - | 70.1 | - | 56.7 | - | 31.6 | 51.1 | - | 56.6 | 58.9 | 54.2 |
| GPTVQ | 2.25 | - | 76.2 | - | 71.9 | - | 43.3 | 67.6 | - | 70.0 | 68.2 | 66.2 |
| **ClusComp** | 2.19 | 76.6 | **77.3** | 75.0 | **72.9** | 40.8 | **43.9** | **78.1** | 55.3 | **73.3** | **68.4** | **69.0** |
| ClusComp | 2.14 | 76.7 | 77.1 | 73.5 | 71.6 | 39.9 | 42.8 | 77.5 | 54.6 | 73.1 | 68.0 | 68.4 |
| RTN | 2.13 | - | 58.4 | - | 32.3 | - | 25.5 | 47.9 | - | 39.4 | 48.9 | 42.1 |
| GPTQ | 2.13 | - | 59.5 | - | 40.2 | - | 27.7 | 57.1 | - | 41.6 | 53.4 | 46.6 |
| GPTVQ | 2.13 | - | 75.2 | - | 68.3 | - | 39.5 | 70.7 | - | 65.7 | **67.5** | 64.5 |
| **ClusComp** | **1.99** | 75.6 | **77.7** | 74.7 | **73.6** | 39.9 | **42.1** | **74.0** | 53.0 | **71.0** | 67.1 | **67.6** |

Table C.4: Zero-shot evaluation of the quantized Llama-3-8B, with baseline results taken from (Huang et al., 2024c). "acc" and "acc_n" mean accuracy and normalized accuracy, respectively. We offer the results of all metrics for a convenient comparison of the follow-up works. But only the highlighted **metrics** (excluding BoolQ) are used to calculate the average accuracy.

| Method | #Bit | PIQA | | ARC-e | | ARC-c | | BoolQ | HellaSwag | | WinoGrande | Avg |
|---|---|---|---|---|---|---|---|---|---|---|---|---|
| | | acc | acc_n | acc | acc_n | acc | acc_n | acc | acc | acc_n | acc | |
| **Llama-3-8B** | 16.00 | 79.9 | - | 80.1 | - | 50.4 | - | - | 60.2 | - | 72.8 | 68.6 |
| RTN | 4.13 | 76.6 | - | 70.1 | - | 45.0 | - | - | 56.8 | - | 71.0 | 63.9 |
| GPTQ | 4.13 | 78.4 | - | 78.8 | - | 47.7 | - | - | 59.0 | - | 72.6 | 67.3 |
| AWQ | 4.13 | **79.1** | - | 79.7 | - | 49.3 | - | - | 59.1 | - | **74.0** | 68.2 |
| SliM-LLM | 4.13 | 78.9 | - | 79.9 | - | 49.4 | - | - | 58.7 | - | 72.6 | 67.9 |
| ClusComp | 4.13 | **79.1** | 80.5 | 80.9 | 79.6 | 49.7 | 54.1 | 81.1 | **59.3** | 78.3 | 72.9 | **68.4** |
| RTN | 3.13 | 62.3 | - | 32.1 | - | 22.5 | - | - | 29.1 | - | 54.7 | 40.2 |
| GPTQ | 3.13 | 74.9 | - | 70.5 | - | 37.7 | - | - | 54.3 | - | 71.1 | 61.7 |
| AWQ | 3.13 | 77.7 | - | 74.0 | - | 43.2 | - | - | 55.1 | - | 72.1 | 64.4 |
| SliM-LLM | 3.13 | 77.8 | - | 73.7 | - | 42.9 | - | - | 55.5 | - | **72.8** | 64.5 |
| RTN | 3.00 | 56.2 | - | 31.1 | - | 20.0 | - | - | 27.5 | - | 53.1 | 35.6 |
| GPTQ | 3.00 | 60.8 | - | 38.8 | - | 22.3 | - | - | 41.8 | - | 60.9 | 44.9 |
| AWQ | 3.00 | 71.9 | - | 66.7 | - | 35.1 | - | - | 50.7 | - | 64.7 | 57.8 |
| QuIP | 3.00 | 76.8 | - | 72.9 | - | 41.0 | - | - | 55.4 | - | 72.5 | 63.7 |
| **ClusComp** | 2.87 | **77.7** | 78.8 | **76.0** | 74.5 | **43.9** | 47.6 | 79.0 | **56.0** | 74.6 | 71.0 | **64.9** |
| **ClusComp** | 2.27 | 70.6 | 71.8 | 63.5 | 57.4 | 31.4 | 35.5 | 74.7 | 49.6 | 66.1 | 67.1 | 56.4 |
| RTN | 2.13 | 53.1 | - | 24.8 | - | 22.1 | - | - | 26.9 | - | 53.1 | 36.0 |
| GPTQ | 2.13 | 53.9 | - | 28.8 | - | 19.9 | - | - | 27.7 | - | 50.5 | 36.2 |
| AWQ | 2.13 | 52.4 | - | 24.2 | - | 21.5 | - | - | 25.6 | - | 50.7 | 34.9 |
| SliM-LLM | 2.13 | 57.1 | - | 35.4 | - | 26.1 | - | - | 28.9 | - | 56.6 | 40.8 |
| PB-LLM | 2.13 | 57.0 | - | 37.8 | - | 17.2 | - | - | 29.8 | - | 52.5 | 38.8 |
| **ClusComp** | 2.14 | **68.0** | 67.1 | **54.7** | 49.0 | **26.4** | 28.8 | 71.5 | **47.0** | 63.0 | **62.4** | **51.7** |
| RTN | 2.00 | 53.1 | - | 24.7 | - | 21.9 | - | - | 25.6 | - | 51.1 | 35.3 |
| GPTQ | 2.00 | 52.8 | - | 25.0 | - | 20.5 | - | - | 26.6 | - | 49.6 | 34.9 |
| AWQ | 2.00 | 55.2 | - | 25.2 | - | 21.3 | - | - | 25.4 | - | 50.4 | 35.5 |
| QuIP | 2.00 | 52.9 | - | 29.0 | - | 21.3 | - | - | 29.2 | - | 51.7 | 36.8 |
| **ClusComp** | 2.01 | **70.1** | 69.6 | **63.3** | 57.7 | **31.9** | 34.2 | 66.6 | **44.4** | 58.0 | **58.4** | **53.6** |

Table C.5: The perplexity of the Llama-3-8B backbone in LLaVA-Next-8B, with baseline results from Huang et al. (2024c).

| Method | Setting | Bit | WikiText2 ↓ | C4 ↓ | PTB ↓ |
|---|---|---|---|---|---|
| - | - | 16.00 | 9.5 | 14.8 | 16.3 |
| GPTQ | w4g128 | 4.13 | **9.5** | 14.8 | 17.1 |
| AWQ | w4g128 | 4.13 | 9.9 | 15.3 | **16.9** |
| ClusComp⁻ | s4n65500 | 4.13 | 9.9 | **13.6** | 17.6 |
| **ClusComp** | s4n65500 | 4.13 | 9.7 | **13.6** | 17.7 |
| GPTQ | w3g128 | 3.13 | 13.0 | 19.5 | 28.4 |
| AWQ | w3g128 | 3.13 | 11.7 | 17.9 | **20.2** |
| ClusComp⁻ | s6n65500 | 2.87 | 14.3 | 16.2 | 31.9 |
| **ClusComp** | s6n65500 | 2.87 | **10.7** | **15.3** | 22.0 |
| GPTQ | w2g128 | 2.13 | 83.7 | 3.1e3 | 2.0e2 |
| AWQ | w2g128 | 2.13 | 1.6e6 | 2.0e6 | 2.2e6 |
| ClusComp⁻ | s8n35000 | 2.14 | 7.7e2 | 6.1e3 | 9.2e2 |
| **ClusComp** | s8n35000 | 2.14 | **14.6** | **21.8** | **27.5** |

# D    NEW RESULTS

## D.1    MORE BASELINES

In this section, we compare ClusComp against SqueezeLLM (Kim et al., 2024) and AdaDim (Heo et al., 2024). As presented in Table D.1, ClusComp consistently achieves lower perplexity than SqueezeLLM, at comparable or even lower bit precision. Similarly, as shown in Table D.2, Clus-Comp outperforms AdaDim on both MMLU and CSR benchmarks.

Table D.1: The perplexity of Llama-2 on WikiText2. The values in the brackets are the exact bits of ClusComp for different LLMs. The SqueezeLLM results are taken from Kim et al. (2024).

| Method | #Bit | Llama-2-7B | Llama-2-13B | Llama-2-70B |
|---|---|---|---|---|
| - | 16.00 | 5.47 | 4.88 | 3.31 |
| SqueezeLLM | 4.27 | 5.57 | 4.96 | - |
| **ClusComp** | $\leq$ **4.14** | **5.54** (4.14) | **4.94** (4.09) | - |
| SqueezeLLM | 3.02 | 6.18 | 5.36 | 3.77 |
| **ClusComp** | $\leq$ **2.89** | **5.86** (2.89) | **5.18** (2.81) | **3.72** (2.72) |
| SqueezeLLM | 2.22 | 10.79 | 7.91 | 4.99 |
| SqueezeLLM | 2.05 | 13.64 | 8.56 | 5.38 |
| SqueezeLLM | 2.01 | 35.49 | 41.02 | 9.44 |
| **ClusComp** | $\leq$ **2.00** | **7.50** (2.00) | **6.17** (1.99) | **4.83** (1.85) |

Table D.2: The accuracy of quantized LLMs on MMLU and four commonsense reasoning (CSR) tasks (PIQA, HellaSwag, WinoGrande and ARC-easy). Following AdaDim, we use lm-eval v0.3.0 (Gao et al., 2024) for the evaluation. The GPTQ-AdaDim results are taken from Heo et al. (2024).

| Method | #Bit | Llama-2-7B | | Llama-2-13B | |
|---|---|---|---|---|---|
| | | MMLU (5-shot ↑) | CSR (0-shot ↑) | MMLU (5-shot ↑) | CSR (0-shot ↑) |
| - | 16.00 | 46.0 | 67.9 | 55.6 | 70.3 |
| GPTQ-AdaDim | 4.13 | 45.3 | 67.7 | 54.6 | 70.1 |
| **ClusComp** | $\leq$ **4.14** | **45.6** | **68.2** | **55.1** | **70.8** |
| GPTQ-AdaDim | 3.13 | 41.3 | 66.4 | **52.3** | 68.7 |
| **ClusComp** | $\leq$ **2.89** | **43.2** | **67.2** | **52.3** | **69.5** |

## D.2    VISUALIZATION OF CLUSCOMP

To illustrate how ClusComp effectively simulates the original weight distribution, we compare it (non-uniform compression) to OmniQuant (uniform quantization) in Figure D.1. The figures demonstrate that ClusComp more closely approximates the 16-bit weight distribution, primarily due to its non-uniform compression approach. Specifically, ClusComp clusters similar vectors over groups of length $g$ across different rows and columns of the weight matrix.

## D.3    QUANTIZATION DIFFICULTY TREND OF LLAMA SERIES

Previous works (Lin et al., 2024; Sun et al., 2024a; Heo et al., 2024) suggest that weight patterns can be identified based on activations rather than relying solely on weight magnitudes. Following a recommendation from Reviewer wLyR at ICLR 2025, we incorporate an additional analysis based on the Wanda score (Sun et al., 2024a) distribution to illustrate the increasing challenges of quantization across the Llama series.

Given the weight matrix $\boldsymbol{W} \in \mathbb{R}^{d_{out} \times d_{in}}$ of a linear layer and the input activations $\boldsymbol{X} \in \mathbb{R}^{NL \times d_{in}}$, where $N$ and $L$ represent the batch and sequence dimensions, the Wanda score $\boldsymbol{S} \in \mathbb{R}^{d_{out} \times d_{in}}$ is computed as $\boldsymbol{S}_{ij} = |\boldsymbol{W}_{ij}| \cdot ||\boldsymbol{X}_j||_2$. A smaller Wanda score within a row of $\boldsymbol{W}$ (on a per-output basis) indicates a less significant weight element.

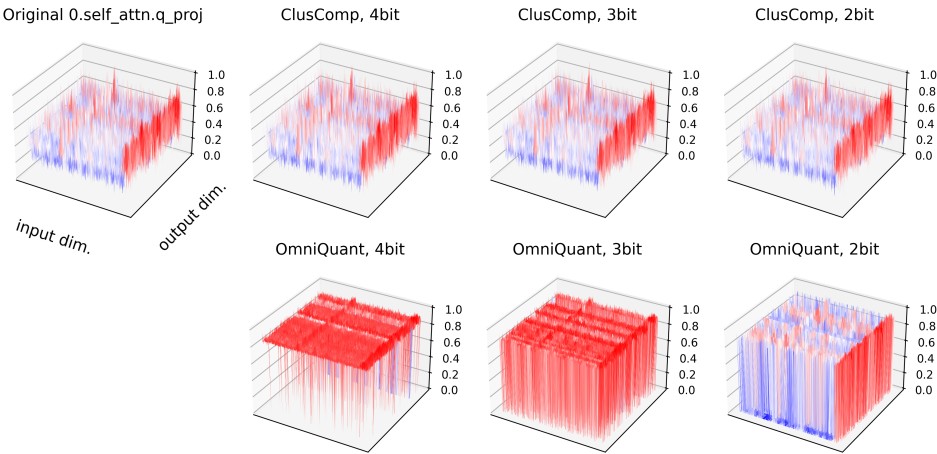

(a) The query projection layer in the first block. Surprisingly, even the 4-bit OmniQuant (uniform quantization) could not simulate the original weight distribution, demonstrating that the first query layer is difficult for quantization. We also observe this for the first key projection layer in Figure D.4(b). However, ClusComp with 4-2 bits perfectly simulates the original weight distribution.

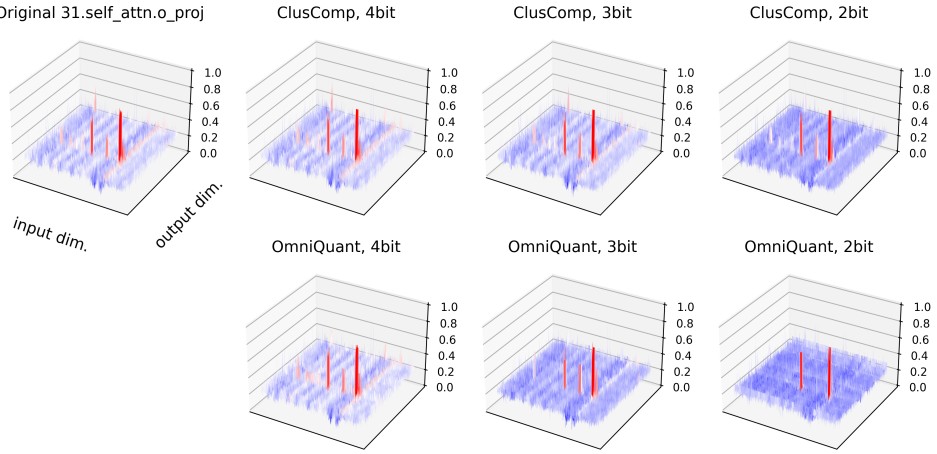

(b) The output projection layer in the last block. Compared to OmniQuant (uniform quantization), ClusComp can better simulate the weight distribution, more evident for the 3 and 2-bit levels.

Figure D.1: Weight patterns of two cherry-picked layers of Llama-2-7B. Darker red and blue indicate larger and smaller weight values, respectively. To make the weight pattern more obvious, we apply these sequential processing steps: (1) take the absolute weight values; (2) downsample the grids with $8 \times 8$ maxpool kernels; (3) calculate the logarithm of these values; (4) normalize the log-values. We also offer the visualization of all layers in the first and last blocks of Llama-2-7B in Figure D.4 and D.5. Overall, ClusComp's weight distribution of different bit-levels can better simulate the original weight distribution.

In Figure D.2 (Right), we present the standard deviation of the Wanda scores across different layers. The results show that Llama-2 exhibits a larger standard deviation compared to Llama-1, while Llama-3 exceeds Llama-2 in this metric. A higher standard deviation reflects a more dispersed Wanda score distribution, indicating that a greater proportion of weight elements are effective and diverse. Consequently, quantization becomes more challenging, as the expanded distribution stretches the quantization grid.

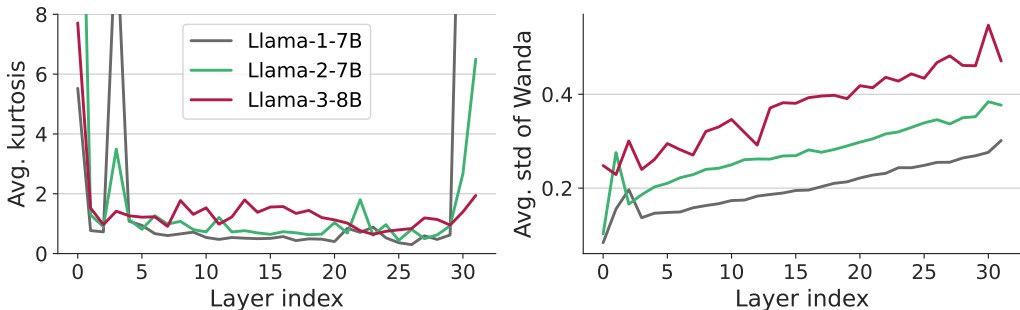

Figure D.2: From Llama-1 to Llama-3, LLMs exhibit increasing challenges for quantization. **Left**: The average kurtosis of weights across various layers in the Llama series, previously shown in Figure 2 (Right). **Right**: The average standard deviation of the Wanda score across various layers. Please refer to Figure D.6 for the Wanda scores of different layer types. Both metrics indicate that Llama-3 has higher variance in most layers, reflecting the presence of more outliers and thus greater difficulty for quantization.

### D.4 QUANTIZATION OF THE CODEBOOK

Thanks to the suggestion of Reviewer h5Zw at ICLR 2025, we conduct further quantization on the codebook $C$. Originally, the data type in the codebook was 16-bit, which facilitates our following recovery training or finetuning step. However, if we can further quantize the codebook, we have two additional advantages: (1) The model size can be further slightly reduced (Only slightly, since the majority of bits is allocated to the code $q$.); (2) The inference can speed up, since the codebook becomes smaller. In sum, we can keep the codebook in 16-bit if we want to do the recovery training or finetuning. If we are only interested in inference, we can further quantize the codebook to a lower bit.

As shown in Table D.3, the performance doesn't change if we quantize the codebook from 16-bit to 8-bit. When quantizing the codebook to 4-bit, the perplexity slightly increases, but is still comparable to the best baseline at the 4-bit level and outperforms the best baseline at the 2-bit level. However, if we further quantize the codebook to the 2-bit level, the perplexity increases significantly. Therefore, we can safely quantize the codebook to 8-bit or 4-bit.

Table D.3: The perplexity of ClusComp with quantized codebook on WikiText2. The results of the best baseline are taken from Table C.1. The values in the brackets are the exact bits for different LLMs. We can observe: (1) 8-bit codebook offers the same perplexity as 16-bit's; (2) 4-bit codebook slightly hurts the performance, but is still comparable to the best baseline at the 4-bit level and outperforms the best baseline at the 2-bit level. (3) The results of the 2-bit codebook are not acceptable.

| Method | Bit for codebook | Avg. Bit | 2-7B | 2-70B | 3-8B | 3-70B |
|---|---|---|---|---|---|---|
| - | - | 16.00 | 5.47 | 3.31 | 6.12 | 2.90 |
| Best baseline | - | 4.13 | 5.58 | 3.39 | 6.50 | 3.30 |
| ClusComp | 16 | ≤ 4.14 | 5.54 (4.14) | 3.40 (4.03) | 6.39 (4.13) | 3.12 (4.03) |
| ClusComp | 8 | ≤ 4.11 | 5.54 (4.11) | 3.40 (4.03) | 6.39 (4.10) | 3.13 (4.03) |
| ClusComp | 4 | ≤ 4.07 | 5.59 (4.07) | 3.43 (4.02) | 6.52 (4.07) | 3.26 (4.02) |
| ClusComp | 2 | ≤ 4.05 | 25.44 (4.05) | 5.64 (4.01) | 1.2e5 (4.05) | 96.52 (4.01) |
| Best baseline | - | ≤ 2.13 | 35.07 (2.00) | 4.64 (2.13) | 85.10 (2.00) | 11.68 (2.13) |
| ClusComp | 16 | ≤ 2.07 | 7.50 (2.00) | 4.37 (2.07) | 12.33 (2.01) | 7.61 (2.07) |
| ClusComp | 8 | ≤ 2.04 | 7.50 (1.92) | 4.37 (2.04) | 12.33 (1.92) | 7.63 (2.04) |
| ClusComp | 4 | ≤ 2.03 | 7.63 (1.86) | 4.42 (2.03) | 12.77 (1.86) | 7.64 (2.03) |
| ClusComp | 2 | ≤ 2.02 | 6.5e3 (1.83) | 21.07 (2.02) | 1.7e5 (1.83) | 2.3e4 (2.02) |

D.5 ABLATION STUDY ON THE GROUP SIZE AND NUMBER OF CLUSTERS

Thanks to the suggestion of Reviewer h5Zw at ICLR 2025, we conducted experiments to determine whether the number of clusters $n$ or the cluster dimension $g$ has a greater impact on the performance of quantized LLMs. As shown in Table D.4, increasing $n$ positively affects performance more than reducing $g$. This finding underpins our choice of $n \approx 2^{16}$. However, while $n$ plays a crucial role in enhancing performance, selecting $n > 2^{16}$ would necessitate using 32-bit storage for the code $q$, substantially increasing the bits-per-parameter and adversely affecting memory efficiency. Therefore, we always choose $n < 2^{16}$.

Table D.4: Ablation study of ClusComp$^-$ on the number of clusters $n$ and the cluster dimension $g$ in the codebook reveals that $n$ plays a more significant role in the performance of the quantized LLM. **(a)** The perplexity remains relatively stable with variations in $g$, although changes in $g$ lead to substantial differences in the bit requirement, as most bits are used to store the codes $q$. Specifically, smaller $g$ values result in larger $q$. Refer to Table 1 for detailed examples. **(b)** In contrast, perplexity is highly sensitive to changes in $n$. Adjusting $n$ causes only a minor change in the bit requirement, as storing the codebook is memory-efficient. **(c)** For comparable bit budgets, $n$ has a greater impact on performance than $g$.

(a) $n = 65500$. Perplexity changes smoothly.

| Setting | #Bit | Wiki2 |
|---------|------|-------|
| g4 | 4.14 | **5.67** |
| g5 | 3.38 | 5.90 |
| g6 | 2.89 | 6.54 |
| g7 | 2.54 | 7.64 |
| g8 | 2.29 | 11.10 |

(b) Same $g$. Perplexity changes dramatically.

| Setting | #Bit | Wiki2 |
|---------|------|-------|
| g7n16384 | 2.35 | **23.14** |
| g7n4096 | 2.30 | 9.4e2 |
| g8n65500 | 2.29 | **11.10** |
| g8n50000 | 2.15 | 21.90 |
| g8n4096 | 2.02 | 5.4e3 |

(c) Similar bit level. Perplexity is more sensitive to $n$.

| Setting | #Bit | Wiki2 |
|---------|------|-------|
| g7n16384 | 2.35 | 23.14 |
| g7n4096 | 2.30 | 9.4e2 |
| g8n65500 | 2.29 | **11.10** |

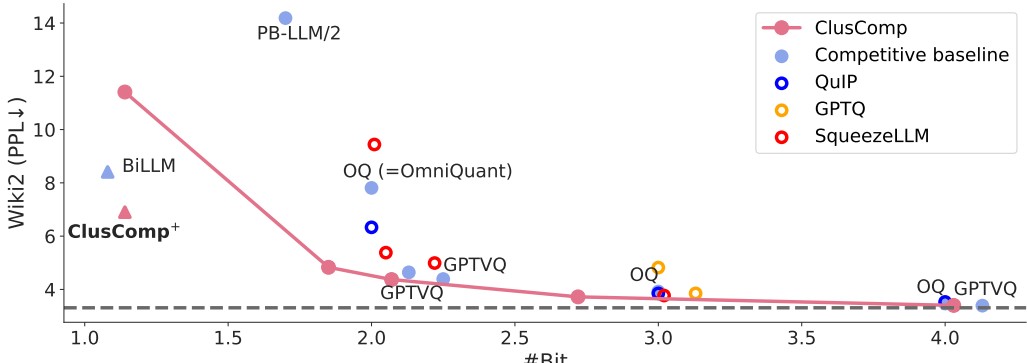

Figure D.3: Perplexity of various methods on Llama-2-70B. Compared to Figure 1, we add three new baselines: GPTQ (Frantar et al., 2022), QuIP (Chee et al., 2023) and SqueezeLLM (Kim et al., 2024). For GPTQ, we only show the results $\geq$ 3-bit, since its perplexity under 3-bit is large.

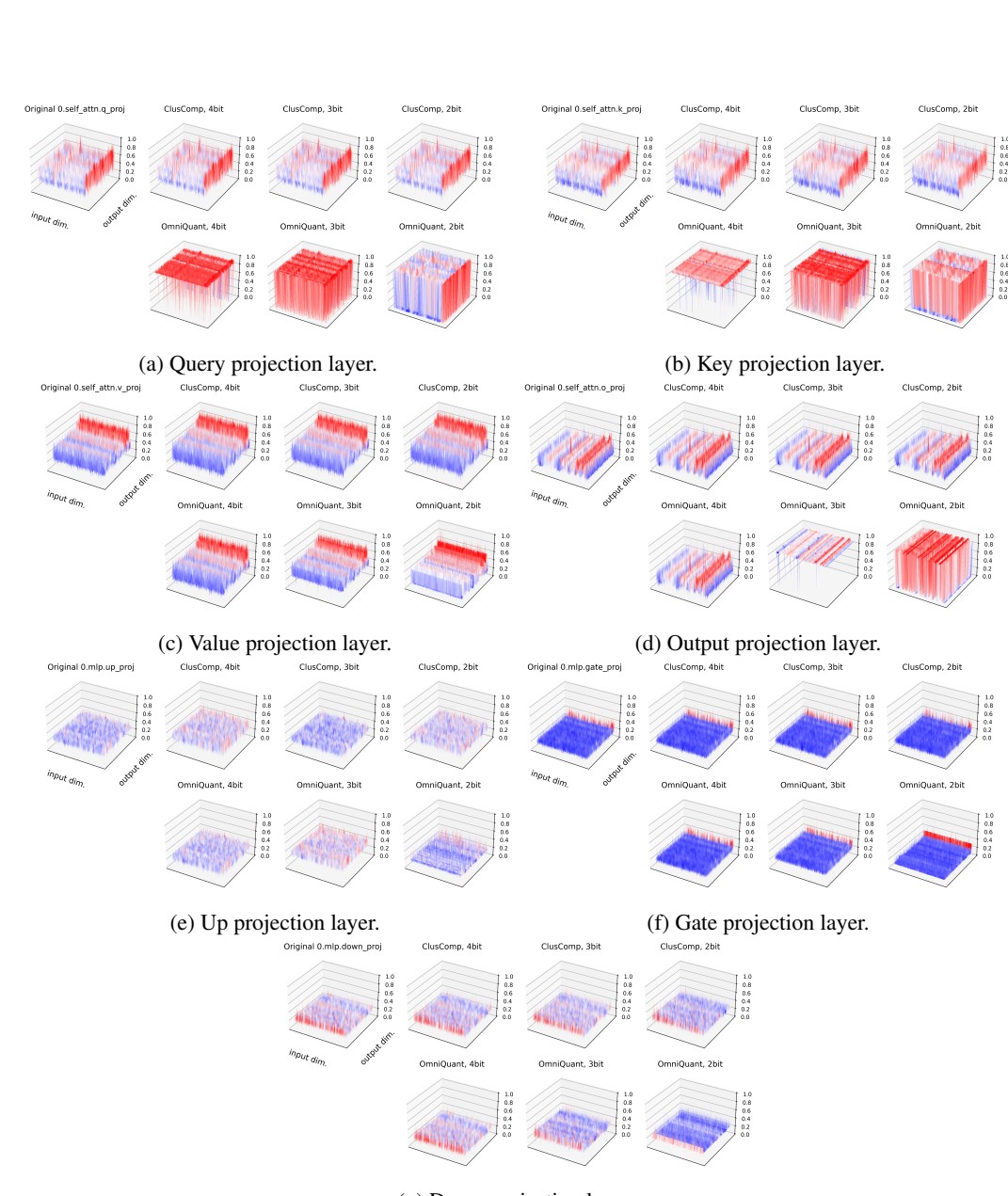

Figure D.4: Weight patterns of all layers in the first block (0-th block) of Llama-2-7B. Darker red and blue indicate larger and smaller weight values, respectively. ClusComp's weight distribution of different bit levels can better simulate the original weight distribution.

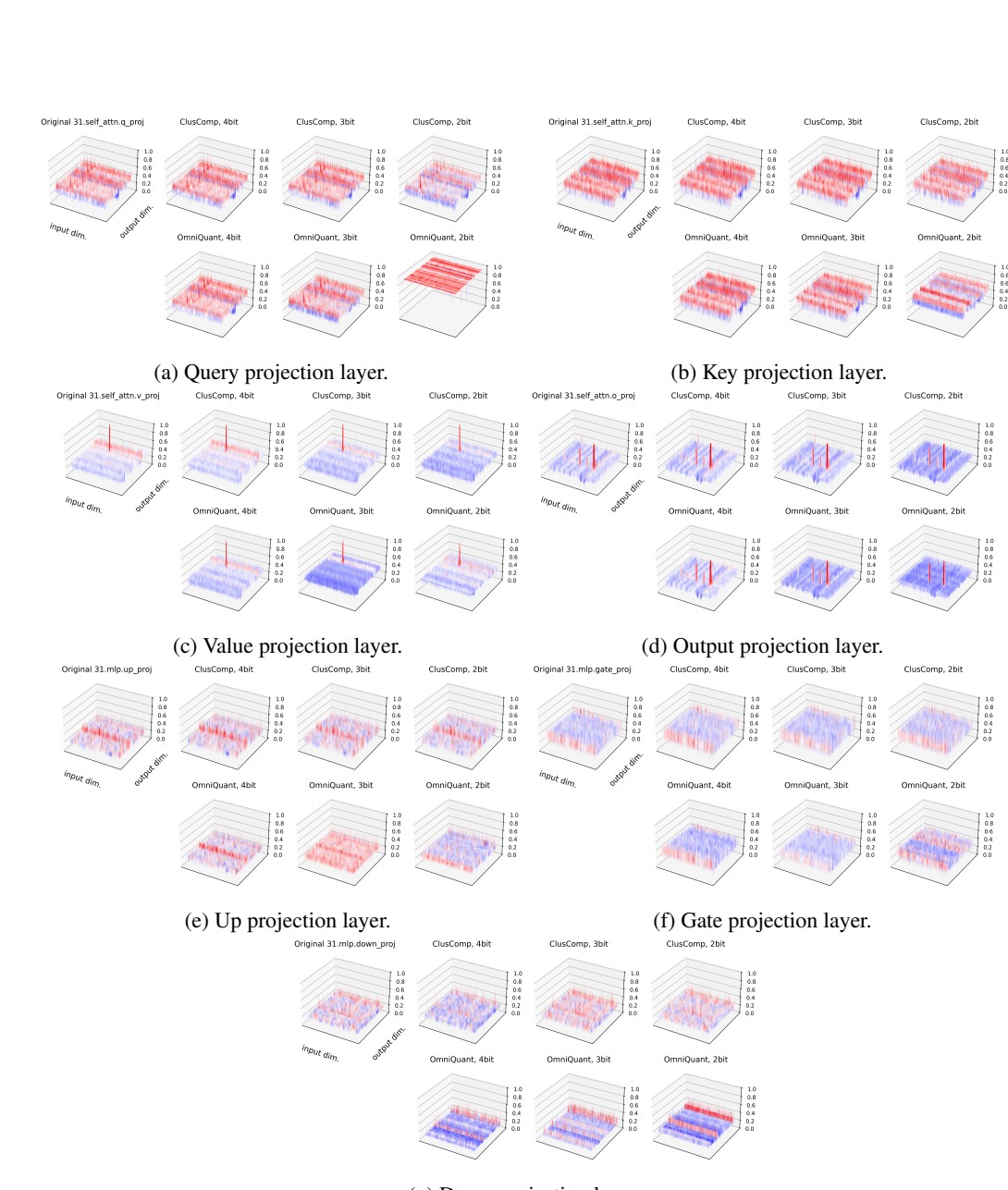

Figure D.5: Weight patterns of all layers in the last block (31-st block) of Llama-2-7B. Darker red and blue indicate larger and smaller weight values, respectively. ClusComp's weight distribution of different bit levels can better simulate the original weight distribution.

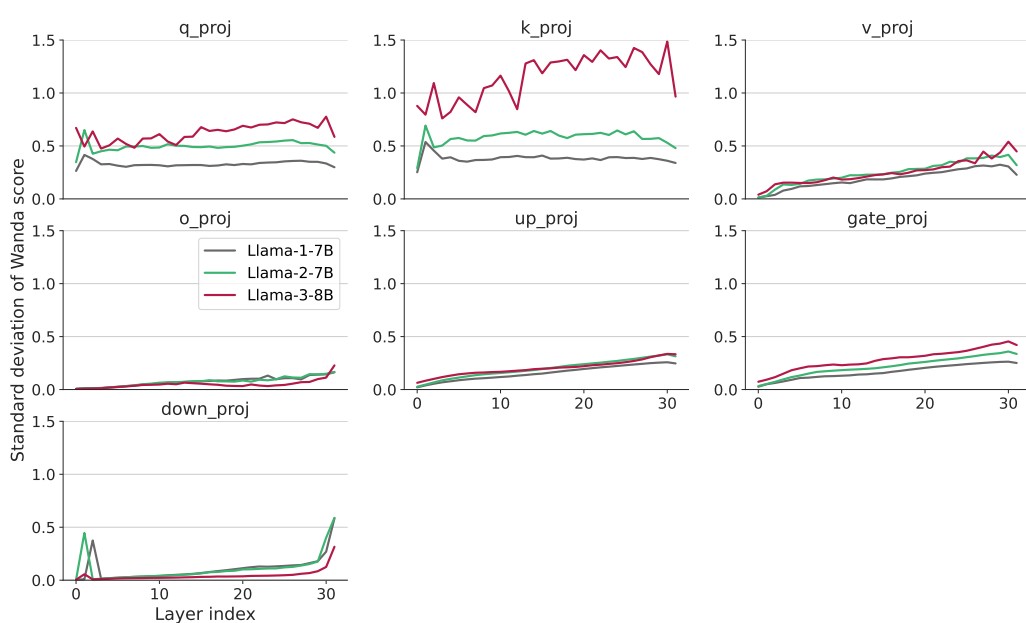

Figure D.6: The standard deviation of Wanda score across various layers in different Llama series reveals three key observations: (1) Deeper layers tend to exhibit higher standard deviation; (2) Three layers (query, key and gate) show a clear trend across Llama series, with Llama-3 showing the highest standard deviation, followed by Llama-2, and then Llama-1; (3) The other four layers show a similar standard deviation for all Llama series.

