# OpenReview forum: "ClusComp: A Simple Paradigm for Model Compression and Efficient Finetuning"
_ICLR.cc/2025/Conference — Submitted to ICLR 2025_

### Official Review · Reviewer_dutm · 2024-10-29

**Soundness:** 4
**Presentation:** 4
**Contribution:** 3
**Rating:** 6
**Confidence:** 5

**Summary:**

The paper presents ClusComp, a way to cluster vectors of values in weights. This post-training 'quantization' technique shows competitive quality, outperforming state-of-the-art quantization methods. The authors evaluate on pre-trained models, multimodal models, and fine-tuned models.

**Strengths:**

1. The paper is well-organized and well-presented. The problem, insight, and method are all clear from reading the abstract alone, and the vast majority of my questions are answered by just reading the paper in detail.
2. Figure 1 is very clear and convincing. We see the obligatory "our line is better than everyone else's", and the ~1 bit model performance, with just a modicum of calibration data, is especially impressive. Out of curiosity, how does SqueezeLLM/GPTQ/QuiP fare, if it was plotted on the same graph?
3. Related work is thorough, and the plots include relevant baselines for comparison -- both popularly-used models and popularly-used compression techniques. This paper includes all relevant baselines that are popularly-used.
4. Evaluation is thorough, with perplexity and zero-shot accuracy results for the pretrained model, as well as downstream multimodal evaluation. On top of that, the authors show lora-tuned results. The results are convincing and clearly presented.

**Weaknesses:**

One missing component is the lack of latency or power benchmarks. In short, this LUT is huge. Per C.1, n could be as large as 65500, which is *basically 2^16. g=5, so the LUT is now 16 * 5 * 2^16 > 2^22 bits or 2^19 bytes. That's 500 KB, which is too large for L1 (H100 has 256 KB of L1, but this needs to fit both the indices AND the LUT), meaning decoding every tile in a matmul could potentially result in many RAM accesses. As a result, I'm curious if a custom kernel for this method would actually be faster than its FP16 baseline, on modern GPUs.

However, future hardware will have increasingly larger and larger SRAMs -- and with it, larger L1s. Given the nature of research, I think this concern can be deprioritized as a result. We'll eventually get to a point where hardware makes this concern obsolete. So, in my opinion, the paper already makes a clear contribution: This is a technique that achieves high quality and would be worth optimizing hardware/kernel performance for. Granted, I don't think it's possible to do in the current form with modern hardware, but maybe its cousin could be made hardware friendly. Papers don't have to be picture perfect out of the gate, for production in the current state or day.

**Questions:**

- What does "time" mean in Figure 1's caption mean? Inference latency? Training time?
- What does $\leq$ mean in Table 2? I looked for mention of this in the main text and caption for Table 2 but couldn't find a mention.

Summary: I don't have much to say here, because the idea was simple, easy to understand, and thoroughly experimented with. Results are shown on pre-trained LLMs, multimiodal LLMs, and fine-tuned LLMs. My main concern is hardware friendliness above, meaning I don't believe latency would actually be reduced due to the size of the look-up-table. However, I still believe this is a paper worth publishing, as it sets a north star for further hardware friendly improvements. There's also a possibility that I'm wrong however. If you can show hardware friendliness for modern hardware, I would bump my rating up!

---

> ### Author Response · Authors · 2024-11-22
> **Thanks for your review**
>
> Dear Reviewer dutm,
>
> Thank you for your thorough review. We are really encouraged by your highlights:
>
> 1. Our paper is **well-organized** and **well-presented**, with **clear problem statement, insight and method**.
> 2. The **$\approx$ 1 bit performance of ClusComp is especially impressive** by only using a modicum calibration data.
> 4. The related works are thorough, and we **compare ClusComp to all relevant and popular-used baselines**.
> 5. We conduct **thorough evaluation** with **convincing and clearly presented results**.
> 6. Our idea is **simple** and **easy to understand**.
> 7. Our paper is **worth publishing**, since **we set a north star for further hardware-friendly improvements**.
>
> $~$
>
> ---
>
> $~$
>
> > **Strength 2. How does SqueezeLLM/GPTQ/QuIP fare, if it was plotted in Figure 1?**
>
> We add these baselines in Figure D.3 in the new submission. One can observe that SqueezeLLM and QuiP are two comparable baselines for >=3 bit. For lower bit levels, ClusComp outperforms them.
>
> In Figure 1, we aim to only include the most competitive baselines. Thanks for your question, we realize that QuIP outperforms OmniQuant at the 2-bit level and have added QuIP to Figure 1.
>
> $~$
>
> > **Weakness: Lack of latency or power benchmarks. If a custom kernel for ClusComp would actually be faster than its FP16 baseline on modern GPUs.**
>
> One of the future plans outlined in our paper was to design a new CUDA kernel optimized for ClusComp (Line 930). We are pleased to share that we successfully implemented this kernel before the rebuttal deadline.
>
> **Summary: The new CUDA kernel speeds up ClusComp's inference, making it > 17% faster than the 16-bit LLM.**
>
> Firstly, we would like to thank you for your thorough review. They are very inspired, and help us come up with two solutions (one is done and the other needs more time) to speed up the ClusComp's inference.
>
> 1. For GPU with small SRAM, like RTX3090-24GB, we mainly optimize the kernel by fusing all operations (index, reshape and matrix multiplication). We can obtain a speedup over the 16-bit LLM, as shown in Table R1. However, we also admit that this speedup mainly comes from the smaller cost of transfer from global GPU memory to cache, mainly working for GPUs whose memory transfer is a severe bottleneck.
>
> 2. For some advanced GPUs (like H100), we notice there is a new feature (distributed shared memory) in the CUDA document that states "Thread block clusters introduced in compute capability 9.0 provide the ability for threads in a thread block cluster to access shared memory of all the participating thread blocks in a cluster." We expect this new feature to offer a fundamental solution for ClusComp. However, due to the time limit and limited access to H100, we couldn't finish this part by now.
>
> We want to re-state that one of the key contributions of ClusComp lies in its ability to deliver significantly better performance at lower bit levels, where uniform quantization often fails. For example, the uniform quantization (GPTQ and AWQ) obtains 0% accuracy on 2-bit LlaVa in Table 3 of the paper, while ClusComp's accuracy is 53.5%. We hope this quality-latency trade-off doesn't hurt our contribution.
>
> $~$
>
> Table R1. Inference speed on RTX3090-24GB GPU. Setup: Generate 128 tokens with a batch size of 1.
> | #Bit | Llama-2-7B | Llama-2-70B |
> | --- | --- | --- |
> | 16 | 1.00x (58.1 tokens/second) | 1.00x (5.2 tokens/second)|
> | 2 | 1.29x | 1.17x |
>
>
>
> $~$
>
> > **Question 1. What does "time" mean in Figure 1?**
>
> Sorry for the confusion. The time (GPU-hour) in Figure 1 means the time required for compression instead of the inference time. ClusComp requires two steps for compression, i.e. clustering and block-wise error minimization. The shown GPU-hour is used for these two steps. The orange bar is for clustering and the green bar is for reconstruction.
>
> $~$
>
> > **What does "≤" mean in Table 2?**
>
> Sorry for this confusion. ClusComp uses a non-uniform quantization technique, which means we can't control the bits-per-parameter to any specific number. Taking "<=4.14" in Table 2 as an example, "<=" means that the largest bits-per-parameter for all LLMs compressed by ClusComp is 4.14. If you go to Table C.1, you can notice that the bits-per-parameter is 4.14 for Llama-1-7B and Llama-2-7B, but it is 4.09 for Llama-2-13B and 4.03 for Llama-2-70B.
>
> $~$
>
> ---
>
> $~$
>
> If you are interested in how ClusComp approximates the 16-bit weight distribution, we highly recommend you to Figure D.1 for the visualization of weight distribution.
>
> Thank you for your thoughtful discussions. We have incorporated the new results in the updated version.
>
> If these revisions address your concerns, we kindly request a reconsideration of the scores. Should you have any further questions, we are happy to assist.

---

> > ### Comment · Reviewer_dutm · 2024-11-27
> > **Thanks for the responses**
> >
> > Props to the authors for actually implementing the kernel and measuring throughput. That’s non-trivial to do!
> >
> > The problem right now is that at bitwidths of >= 3bpw you’re at similar accuracy but much lower throughput. So cluscomp is strictly worse.
> >
> > My hot take would be to focus 100% on the 1bpw model — no one wants a 0-accuracy model, no matter now fast. Then, benchmark 1bpw cluscomp kernel and at a bare minimum, show that the throughput hit is tolerable.
> >
> > (Ideally, you’d show sota throughout vs quality trade off in a fig1, because bpw doesn’t tell the whole story. I think this should be the focus post rebuttals leading up to your final submission. I already think this is worthy of publishing, and the consensus after rebuttals looks like an accept.)
> >
> > Time and <= makes sense now.

---

> ### Author Response · Authors · 2024-11-27
> **Thank you very much for your suggestion**
>
> Dear Reviewer dutm,
>
> We are really encouraged by your further confirmation that **our paper is worthy of publishing**, and thank you for your suggestion of focusing more on low-bit post rebuttal.
>
> In the following, we want to highlight our other contributions and new latency results that you are interested.
>
> 1. **ClusComp as parameter and memory-efficient finetuning method**: Since most of the discussion is about the post-training quantization performance of ClusComp, we want to metion our another core contribution: **ClusComp can also serve as a memory-efficient and parameter-efficient finetuning method that outperform LoRA, QLoRA, GaLore and LISA.. One can finetune a 70B LLM on a GPU with mere 48GB memory, and achieve promising accuracy as full finetuning.**
>
> 2. Thanks to the suggestion from Reviewer h5Zw who recommends us to **further quantize the codebook from 16-bit (originally) to lower bit**. I.e. after the block-wise error minimization, we further quantize each cluster in the codebook with symmetric quantization. (Notably, we quantize the codebook mainly for faster inference. If one wants to do finetuning, we still need to save it in FP16 for finetuning, and then quantize them for faster inference.)
>
> From Table R2, we can observe: (1) 8-bit codebook offers the same perplexity as 16-bit’s; (2) 4-bit codebook slightly hurts the performance, but is still comparable to the best baseline at the 4-bit level and significantly outperforms the best baseline at the 2-bit level; (3) The results of the 2-bit codebook are not acceptable.
>
> Since the quantization further reduces the codebook size, we can obtain better inference speed. As shown in Table R3, we obtain **40% speedup for a 7B LLM and 22% for a 70B LLM**.  Notably, our kernel is mainly optimized for the 16-bit codebook. We expect a higher speedup with further optimization, and will continue this work.
>
> $~$
>
> Table R2. The perplexity of ClusComp with quantized codebook on WikiText2. The best baseline is taken from Table C.1 in the paper. Please refer to Table D.3 in the paper for better reading experience and more results (70B LLM offers the same observation).
> | Method | Bit for codebook | Avg. Bit | Llama-2-7B | Llama-3-8B |
> | --- | --- | --- | --- | --- |
> | - | - | 16.00 | 5.47 | 6.12 |
> | Best baseline | - | 4.13 | 5.58 | 6.50 |
> | ClusComp | 16 | 4.14 | 5.54 | 6.39 |
> | ClusComp | 8 | 4.11 | 5.54 | 6.39 |
> | ClusComp | 4 | 4.07 | 5.59 | 6.52 |
> | ClusComp | 2 | 4.05 | 25.44 | 1.2e5 |
> |-|
> | Best baseline | - | 2.00 | 35.07 | 85.10 |
> | ClusComp | 16 | 2.01 | 7.50 | 12.33 |
> | ClusComp | 8 | 1.92 | 7.50 | 12.33 |
> | ClusComp | 4 | 1.86 | 7.63 | 12.77 |
> | ClusComp | 2 | 1.83 | 6.5e3 | 1.7e5 |
>
> Table R3. Inference speed on RTX3090-24GB GPU.
> | Bit for codebook | Avg. Bit | Llama-2-7B | Llama-2-70B |
> | --- | --- | --- | --- |
> | - | 16 | 1.00x | 1.00x |
> | 16 | 2.01 | 1.29x | 1.17x |
> | 8 | 1.92 | 1.31x | 1.19x |
> | 4 | 1.86 | 1.40x | 1.22x |

---

> ### Author Response · Authors · 2024-12-01
> **≈ 100% inference speedup**
>
> Dear Reviewer dutm,
>
> thanks to the extension of the rebuttal deadline and your suggestion of focusing more on low-bit optimization, we could finally achieve a **≈ 100% inference speedup** after carefully optimizing the CUDA kernel. Please refer to the [general response](https://openreview.net/forum?id=YZEzVR5awV&noteId=30x15AjS0D) for more details.
>
> If this work further addresses your concern, and it is not too much trouble for you, we would really thank you for any update to the score.

---

### Official Review · Reviewer_ZEQz · 2024-10-31

**Soundness:** 3
**Presentation:** 3
**Contribution:** 3
**Rating:** 6
**Confidence:** 4

**Summary:**

The paper proposes a simple clustering based framework to quantize LLMs and finetune them. The authors employ k-means clustering to find the clustering centroids (i.e., the codebook) over small sub-vectors of the weight matrix. For finetuning, the paper proposes to finetune the entire codebook as opposed to QLoRA [1] which only finetunes the LoRA parameters while keeping the backbone quantized LLM frozen. For weight-only quantization and parameter-efficient finetuning, the paper improves over most baselines.

**Strengths:**

- A simple alternative to scalar quantization, especially in scenarios where the size of the LLM is a bigger bottleneck than its latency.
- An efficient strategy to finetuning entire LLMs.

**Weaknesses:**

- The proposed approach doesn't give latency benefits. However, the authors acknowledge this point.
- Some works that use codebooks and non-uniform quantization are missing from the related works section: SqueezeLLM: Dense-and-Sparse Quantization [2], Any-Precision LLM: Low-Cost Deployment of Multiple, Different-Sized LLMs [3].

**Questions:**

- Does finetuning the codebook affect the generalization performance of the model? In the case of QLoRA, since the backbone LLM is frozen, the generalization properties of the LLMs are preserved.



[1] Dettmers, Tim et al. “QLoRA: Efficient Finetuning of Quantized LLMs.” NeurIPS 2023.

[2] Kim, Sehoon et al. “SqueezeLLM: Dense-and-Sparse Quantization.” ICML 2024.

[3] Park, Yeonhong et al. “Any-Precision LLM: Low-Cost Deployment of Multiple, Different-Sized LLMs.” ICML 2024.

---

> ### Author Response · Authors · 2024-11-22
> **Thanks for your review**
>
> Dear Reviewer ZEQz,
>
> Thank you for your thorough review. We are really encouraged by your highlights:
>
> 1. Our proposed method, ClusComp, is a **simple** alternative to scalar quantization, and **suitable for scenarios where memory is the bottleneck** instead of latency.
> 2. ClusComp can serve as **an efficient finetuning method** to train the entire LLM.
>
> $~$
>
> ---
>
> $~$
>
> > **Weakness 1. ClusComp doesn't give latency benefits.**
>
> One of the future plans outlined in our paper was to design a new CUDA kernel optimized for ClusComp (Line 930). We are pleased to share that we successfully implemented this kernel before the rebuttal deadline.
>
> **Summary: The new CUDA kernel enhances ClusComp's inference speed, achieving a >17% improvement compared to the 16-bit LLM.**
>
> In the initial submission, we had not yet developed this kernel, and the inference speed of ClusComp was comparable to (or slightly slower than) the 16-bit LLM. After submission, we focused on designing an efficient CUDA kernel for ClusComp, integrating all necessary operations (indexing, reshaping, and matrix multiplication). As shown in Table R1, our preliminary kernel implementation now improves inference speed by over 17%.
>
> While the speedup remains lower than that of uniform quantization methods, this is expected, as uniform quantization is inherently more GPU-friendly. However, one of the key contributions of ClusComp lies in its ability to deliver significantly better performance at lower bit levels, where uniform quantization often fails. For instance, in Table 3, we show that uniform quantization methods (e.g., GPTQ and AWQ) achieve 0% accuracy on 2-bit LLaVA, whereas ClusComp attains an impressive 53.5% accuracy.
>
>
> $~$
>
> Table R1. Inference speed on RTX3090-24GB GPU. Setup: Generate 128 tokens with a batch size of 1.
> | #Bit | Llama-2-7B | Llama-2-70B |
> | --- | --- | --- |
> | 16 | 1.00x (58.1 tokens/second) | 1.00x (5.2 tokens/second)|
> | 2 | 1.29x | 1.17x |
>
> $~$
>
>
> > **Weakness 2. Add missing related works, like SqueezeLLM [2] and Any-Precision LLM [3].**
>
> These two works are indeed very related. We have added them at Line-61 and Line-124 in the new submission.
>
> $~$
>
> > **Question. Does finetuning the codebook affect the generalization performance of the model?**
>
> This is a good point. **In contrast, our finetuning results on the general-domain tasks demonstrate a better generalization of ClusComp**.
>
> In Table 6 and 7, we finetune the LLMs on Alpaca-GPT3.5 and Alpaca-GPT4, and evaluate the finetuned LLMs on MMLU and AGIEval. Alpaca contains 52K instruction-following samples. MMLU and AGIEval is a large collection of varied tasks. Specifically, AGIEval contains 20 tasks that focus on history, math, biology, law, logic, chemistry, English and so on. MMLU has a similar focus with even more tasks. We can safely assume that the data distribution in Alpaca is not fully aligned with that in MMLU and AGIEval. So overfit to Alpaca doesn't guarantee a better performance on these benchmarks.
>
> We confirm this by finetuning Llama-2-7B compressed by ClusComp with one more epoch. (Notably, our reported results in the paper strictly follow the number of epochs as our baselines). The results become worse, from 47.0 to 46.4 on MMLU and from 26.5 to 26.1 on AGIEval.
>
> ClusComp's better results on MMLU and AGIEval against quantization-related finetuning methods (QA-LoRA, GPTQ-LoRA) and memory-efficient finetuning methods (LISA, LoRA, GaLore) actually show that ClusComp maintains a good generalization, since these two large benchmarks couldn't easily be fit.
>
> We further show the weight distribution of ClusComp in Figure D.1 of the new submission, one can observe that ClusComp closely approximates the 16-bit weight distribution. This is a major reason for ClusComp's better performance since it better preserves the pretrained knowledge.
>
> $~$
>
> ---
>
> $~$
>
> Thank you for your thoughtful questions. We have incorporated the new results in the updated version.
>
> If these revisions address your concerns, we kindly request a reconsideration of the scores. Should you have any further questions, we are happy to assist.

---

> > ### Comment · Reviewer_ZEQz · 2024-11-24
> >
> > I thank the authors for the detailed response. I would like to keep my score.

---

> > > ### Author Response · Authors · 2024-11-24
> > >
> > > We stilll thank you very much for your thorough review, especially the missing related works and the discussion about generalization. We believe adding these makes our work more solid.

---

> > > ### Author Response · Authors · 2024-12-01
> > > **≈ 100% inference speedup**
> > >
> > > Dear Reviewer ZEQz,
> > >
> > > thanks to the extension of the rebuttal deadline, we could finally achieve a **≈ 100% inference speedup** after carefully optimizing the CUDA kernel. Please refer to the [general response](https://openreview.net/forum?id=YZEzVR5awV&noteId=30x15AjS0D) for more details.
> > >
> > > If this work further addresses your concern, and it is not too much trouble for you, we would really thank you for any update to the score.

---

### Official Review · Reviewer_h5Zw · 2024-11-02

**Soundness:** 2
**Presentation:** 2
**Contribution:** 2
**Rating:** 5
**Confidence:** 4

**Summary:**

The paper introduces ClusComp, a clustering-based paradigm for compressing large language models (LLMs). It flattens and reshapes the weight matrix and clustering the segments to a set of centroids while not reduces the bits of weights. It claims significant advancements in compression efficiency without performance degradation, even reaching 1-bit precision with improved finetuning outcomes over existing method.

**Strengths:**

1. **Empirical Results**: ClusComp achieves superior performance at low-bit precisions, outperforming alternative methods across various LLMs and multimodal models. Its efficacy is well-supported by extensive experiments.

2. **Memory Efficiency**: ClusComp demonstrates significant memory savings, making it feasible to deploy large models on resource-constrained devices, which is crucial for practical deployment.

**Weaknesses:**

1. ClusComp does not reduce the parameter count through dimensional reduction or bit-width reduction, thus maintaining the same computational load during inference.

2. Some visualizations lack clarity:
    - _Figure 1_: The meaning of the marker colors is unclear.
    - _Figure 3_: The histogram appears unnecessary; a simple text explanation could suffice.

3. The block-wise error minimization introduced in section 3.2.3 is a standard practice in post-training quantization and lacks novelty in this context.

**Questions:**

1. _Line 253-254_: The authors state that the codebooks are not quantized. Could further quantization improve the efficiency of the model? Would combining ClusComp with conventional quantization lead to better results?

2. _Figure 4 and Table C.4_: Why does the 2.01-bit ClusComp version (53.6% accuracy) perform better than the 2.14-bit version (51.7%) on Llama3-8B?

3. Could additional ablation studies on hyperparameters, such as cluster count and codebook dimensionality, provide more insight into their impact on compression performance?

---

> ### Author Response · Authors · 2024-11-22
> **Thanks for your review (Response 1/n)**
>
> Dear Reviewer h5Zw,
>
> Thank you for your thorough review. We are really encouraged by your highlights:
>
> 1. Our proposed method, ClusComp, achieves **superior performance at low-bit precisions**, outperforming alternative methods **acrosss various LLMs and VLM** with **extensive experiments**.
> 2. ClusComp demonstrates **significant memory savings**, making it feasible to deploy LLMs on resource-constrained devices, and crucial for practical deployment.
>
> $~$
>
> ---
>
> $~$
>
> > **Weakness 1. ClusComp maintains the same computational load during inference as the FP16 version.**
>
> One of the future plans outlined in our paper was to design a new CUDA kernel optimized for ClusComp (Line 930). We are pleased to share that we successfully implemented this kernel before the rebuttal deadline.
>
> **Summary: The new CUDA kernel enhances ClusComp's inference speed, achieving a 29% speedup for a 7B LLM and a 17% speedup for a 70B LLM.**
>
> In the initial submission, we had not yet developed this kernel, and the inference speed of ClusComp was comparable to (or slightly slower than) the 16-bit LLM. After submission, we focused on designing an efficient CUDA kernel for ClusComp, integrating all necessary operations (indexing, reshaping, and matrix multiplication). As shown in Table R1, our preliminary kernel implementation now improves inference speed by > 17%.
>
> While the speedup remains lower than that of uniform quantization methods, this is expected, as uniform quantization is inherently more GPU-friendly. However, one of the key contributions of ClusComp lies in its ability to deliver significantly better performance at lower bit levels, where uniform quantization often fails. For instance, in Table 3, we show that uniform quantization methods (e.g., GPTQ and AWQ) achieve 0% accuracy on 2-bit LLaVA, whereas ClusComp attains an impressive 53.5% accuracy.
>
> $~$
>
> Table R1. Inference speed on RTX3090-24GB GPU. Setup: Generate 128 tokens with a batch size of 1.
> | #Bit | Llama-2-7B | Llama-2-70B |
> | --- | --- | --- |
> | 16 | 1.00x (58.1 tokens/second) | 1.00x (5.2 tokens/second)|
> | 2 | 1.29x | 1.17x |
>
> $~$
>
> > **Weakness 2. What is the meaning of the marker colors in Figure 1.**
>
> We use different colors to distinguish the baselines from ClusComp, i.e. using blue color for all baselines and using red color for ClusComp. Sorry for this confusion, we have added a legend in the new submission for Figure 1.
>
> $~$
>
> > **Weakness 2. The histogram in Figure 3 appears unnecessary; a simple text explanation could suffice.**
>
> Thank you for this suggestion. We try to support our claim with empirical observation. We will move it to the appendix for our next version.
>
> $~$
>
> > **Weakness 3. Block-wise error minimization lacks novelty.**
>
> We acknowledge that block-wise error minimization is not novel and is a widely adopted approach for reducing quantization error, as stated in L277-278: "Block-wise error minimization... has emerged as a standard, efficient, and effective approach... (Egiazarian et al., 2024; Tseng et al., 2024; van Baalen et al., 2024; Shao et al., 2024; Liao & Monz, 2024a)".
>
> Our intent in making this explicit was to clarify that we do not claim to have introduced this technique. Rather, it serves as an intermediate step to enhance the quality of ClusComp, consistent with practices in prior works such as AQLM, Quip#, GPTVQ, OmniQuant, ApiQ, and others. These methods also employ block-wise error minimization to mitigate quantization error.
>
> The core contributions of our work lie elsewhere. Specifically, we propose a simple yet effective model compression method based on clustering, which surpasses existing baselines across multiple bit levels. Moreover, our method doubles as a parameter-efficient fine-tuning approach (requiring only 1% of trainable parameters) and a memory-efficient fine-tuning technique (reducing memory usage for a 70B LLM from >140GB to 42GB), achieving superior results compared to prior parameter and memory-efficient fine-tuning methods.
>
> $~$
>
> (1/n)

---

> ### Author Response · Authors · 2024-11-22
> **Response n/n**
>
> Following response (1/n)
>
> $~$
>
> > **Question 1. Could further quantization of the codebook improve the efficiency of the model? Would combining ClusComp with conventional quantization lead to better results?**
>
> This is a very inspired suggestion. We have added these new results to Appendix D.4 in the new submission.
>
> **Summary: 8-bit or 4-bit codebook offers a similar perplexity as 16-bit codebook, and they further speed up the inference.**
>
> Experiment setup: We apply symmetric quantization to the codebook parameters after the block-wise error minimization step.
>
> From Table R2, we can observe: (1) 8-bit codebook offers the same perplexity as 16-bit’s; (2) 4-bit codebook slightly hurts the performance, but is still comparable to the best baseline at the 4-bit level and outperforms the best baseline at the 2-bit level. (3) The results of the 2-bit codebook are not acceptable.
>
> Since the quantization further reduces the codebook size, we can obtain better inference speed, as shown in Table R3. Notably, our kernel is mainly optimized for the 16-bit codebook. We expect a higher speedup with further optimization. Due to the time limit, we couldn't finish it by now.
>
> $~$
>
> Table R2. The perplexity of ClusComp with quantized codebook on WikiText2. The best baseline is taken from Table C.1 in the paper. Please refer to Table D.3 in the paper for better reading experience and more results.
> | Method | Bit for codebook | Avg. Bit | Llama-2-7B | Llama-3-8B |
> | --- | --- | --- | --- | --- |
> | - | - | 16.00 | 5.47 | 6.12 |
> | Best baseline | - | 4.13 | 5.58 | 6.50 |
> | ClusComp | 16 | 4.14 | 5.54 | 6.39 |
> | ClusComp | 8 | 4.11 | 5.54 | 6.39 |
> | ClusComp | 4 | 4.07 | 5.59 | 6.52 |
> | ClusComp | 2 | 4.05 | 25.44 | 1.2e5 |
> |-|
> | Best baseline | - | 2.00 | 35.07 | 85.10 |
> | ClusComp | 16 | 2.01 | 7.50 | 12.33 |
> | ClusComp | 8 | 1.92 | 7.50 | 12.33 |
> | ClusComp | 4 | 1.86 | 7.63 | 12.77 |
> | ClusComp | 2 | 1.83 | 6.5e3 | 1.7e5 |
>
> Table R3. Inference speed on RTX3090-24GB GPU.
> | Bit for codebook | Avg. Bit | Llama-2-7B | Llama-2-70B |
> | --- | --- | --- | --- |
> | - | 16 | 1.00x | 1.00x |
> | 16 | 2.01 | 1.29x | 1.17x |
> | 8 | 1.92 | 1.31x | 1.19x |
> | 4 | 1.86 | 1.40x | 1.22x |
>
> $~$
>
> > **Question 2. Why does the 2.01-bit ClusComp version perform better than the 2.14-bit version on Llama3-8B?**
>
> This is an insightful observation. We believe it can be attributed to two key factors:
>
> 1. Unlike perplexity, accuracy is a more discrete and inherently noisier metric. In Table C.4, for instance, 4-bit AWQ notably outperforms the 16-bit Llama-3-8B on WinoGrande (74.0 vs. 72.8). Given that the differences between 2.01-bit and 2.14-bit are small, coupled with the larger performance variance typically observed in low-bit LLMs (as seen in lm-eval), the slightly higher accuracy for the 2.01-bit configuration could stem from these factors.
>
> 2. As shown in Figure D.1 of the updated submission, ClusComp maintains a weight distribution at the 2-bit level that closely resembles the distribution at 16-bit precision. This suggests that a minor reduction in bit level does not necessarily result in performance degradation.
>
>
> $~$
>
> > **Question 3. Additional ablation studies on hyperparameters, such as the number of clusters ($n$) and the cluster dimension ($g$).**
>
> **Summary: The performance is more sensitive to the number of clusters than the cluster dimension.**
>
> We vary $g$ and $n$ on Llama-2-7B with ClusComp$^-$ and show the perplexity in Table R4, R5 and R6. Increasing $n$ has a more positive effect on performance compared to reducing $g$. This finding underpins our choice of $n \approx 2^{16}$ (like 65500). However, while $n$ plays a crucial role in enhancing performance, selecting $n > 2^{16}$ would necessitate using 32-bit storage for the code **$q$**, substantially increasing the bits-per-parameter and adversely affecting memory efficiency. So we always choose $n<2^{16}$. We have added these results to Appendix D.5 in the new submission.
>
> $~$
>
> Table R4. Same $n=65500$ different $g$. Perplexity changes smoothly. Please refer to Table D.4 for better reading experience.
> | g | #Bit | Wiki2 |
> | --- | --- | --- |
> | 4 | 4.14 | 5.67 |
> | 5 | 3.38 | 5.90 |
> | 6 | 2.89 | 6.54 |
> | 7 | 2.54 | 7.64 |
> | 8 | 2.29 | 11.10 |
>
> Table R5. Same $g$ different $n$. Perplexity changes dramatically.
> | Setting | #Bit | Wiki2 |
> | --- | --- | --- |
> | g7n16384 | 2.35 | 23.14 |
> | g7n4096 | 2.30 | 9.4e2 |
> | - |
> | g8n65500 | 2.29 | 11.10 |
> | g8n50000 | 2.15 | 21.90 |
> | g8n4096 | 2.02 | 5.4e3 |
>
> Table R6. Similar bit level. Perplexity is more sensitive to $n$.
> | Setting | #Bit | Wiki2 |
> | --- | --- | --- |
> | g7n16384 | 2.35 | 23.14 |
> | g7n4096 | 2.30 | 9.4e2 |
> | g8n65500 | 2.29 | 11.10 |
>
>
> $~$
>
> ---
>
> $~$
>
> Thank you for your thoughtful suggestions. We have incorporated the new results in the updated version.
>
> If these revisions address your concerns, we kindly request a reconsideration of the scores. Should you have any further questions, we are happy to assist.

---

> > ### Comment · Reviewer_h5Zw · 2024-11-26
> >
> > Thank you for your detailed and constructive rebuttal. I appreciate the effort in addressing reviewers' concerns and providing new insights. The rebuttal effectively addresses most of my concerns and adds meaningful contributions to the manuscript. I will revise my rating to 5.

---

> > > ### Author Response · Authors · 2024-12-01
> > > **≈ 100% inference speedup**
> > >
> > > Dear Reviewer h5Zw,
> > >
> > > thanks to the extension of the rebuttal deadline, we could finally achieve a **≈ 100% inference speedup** after carefully optimizing the CUDA kernel. Please refer to the [general response](https://openreview.net/forum?id=YZEzVR5awV&noteId=30x15AjS0D) for more details.
> > >
> > > If this work further addresses your concern, and it is not too much trouble for you, we would really thank you for any update to the score.

---

> ### Author Response · Authors · 2024-11-27
> **Thank you very much for your increased score**
>
> Dear Reviewer h5Zw,
>
> We sincerely thank you for your positive feedback and the time you dedicated to reviewing our rebuttal. It brings us great joy to learn that our response has addressed your concerns and contributed to increasing the score from 3 to 5.
>
> As the score is still borderline reject, we are wondering if there are any major concerns regarding our current revision. It would be our great pleasure to provide further clarifications and results to address any additional doubts.
>
> Your suggestions really help a lot to improve our work and make the justification of our method more complete. We also greatly appreciate your recognition of the superior performance at low-bit precisions and significant memory savings of our algorithm.
>
> Once again, we would like to express our appreciation for your valuable comments during the reviewing process.
>
> Best regards,
>
> Authors

---

### Official Review · Reviewer_wLyR · 2024-11-03

**Soundness:** 3
**Presentation:** 3
**Contribution:** 3
**Rating:** 6
**Confidence:** 5

**Summary:**

This paper tackles memory savings for LLMs by using clustering-based quantization. ClusComp clusters the weight matrices
to generate codebooks, tunes these codebooks block-by-block to reconstruct intermediate activations, and also conduct recovery training for high compression regimes. It shows promising results for language modeling and also multimodal models across various tasks and model size.

**Strengths:**

- The motivation and method is sound and clear. It is nice to see that they clarify no gains in speedup while highlighting the memory savings and limited accuracy loss.
- Paper presentation and writing are readable and easy to understand.
- Experimental results are promising with a wide range of models, tasks, and comparisons to other methods.

**Weaknesses:**

- I'd like to see more comparison to recent works, including clustering-based quantization [1] and for uniform PTQ [2].
- Some analysis that explains how the codebooks are distributed to better capture salient weights would be helpful to understand why this method works.

[1] Kim, et al., "SqueezeLLM: Dense-and-Sparse Quantization", ICML 2024. \
[2] Heo, et al., "Rethinking Channel Dimensions to Isolate Outliers for Low-bit Weight Quantization of Large Language Models", ICLR 2024

**Questions:**

1. Figure 1: L45, "Some results are divided by a factor for better visualization." -> what does divided by a factor mean? Also, it is unclear what the triangular marker means on the bottom left graph (everything else is a circular marker; what is the difference?).
2. Add to the Figure 2 description that it's showing the kurtosis of weights. Also, recent works [1,2,3] show that activation-awareness is important for identifying weight outliers, so simply taking the weight distribution to explain the quantization difficulty of recent LLMs may be insufficient.

[1] Lin, et al., "AWQ: Activation-aware Weight Quantization for LLM Compression and Acceleration", MLSys 2024.\
[2] Heo, et al., "Rethinking Channel Dimensions to Isolate Outliers for Low-bit Weight Quantization of Large Language Models", ICLR 2024\
[3] Sun, et al., "A Simple and Effective Pruning Approach for Large Language Models", ICLR 2024

---

> ### Author Response · Authors · 2024-11-22
> **Thanks for your review (Response 1/n)**
>
> Dear Reviewer wLyR,
>
> Thank you for your thorough review. We are really encouraged by your highlights:
>
> 1. The motivation and method are **sound and clear**. Our paper is **easy to understand**.
> 2. Our proposed method, ClusComp, **saves memory** and has **limited accuracy loss** compared to the 16-bit LLM.
> 4. ClusComp shows **promising results with a wide range of models, tasks and comparisons.**
>
> $~$
>
> ---
>
> $~$
>
> > **New results on the inference speed of ClusComp.**
>
> One of the future plans outlined in our paper was to design a new CUDA kernel optimized for ClusComp (Line 930). We are pleased to share that we successfully implemented this kernel before the rebuttal deadline.
>
> **Summary: The new CUDA kernel enhances ClusComp's inference speed, achieving a >17% improvement compared to the 16-bit LLM.**
>
> In the initial submission, we had not yet developed this kernel, and the inference speed of ClusComp was comparable to (or slightly slower than) the 16-bit LLM. After submission, we focused on designing an efficient CUDA kernel for ClusComp, integrating all necessary operations (indexing, reshaping, and matrix multiplication). As shown in Table R1, our preliminary kernel implementation now improves inference speed by over 17%.
>
> While the speedup remains lower than that of uniform quantization methods, this is expected, as uniform quantization is inherently more GPU-friendly. However, one of the key contributions of ClusComp lies in its ability to deliver significantly better performance at lower bit levels, where uniform quantization often fails. For instance, in Table 3, we show that uniform quantization methods (e.g., GPTQ and AWQ) achieve 0% accuracy on 2-bit LLaVA, whereas ClusComp attains an impressive 53.5% accuracy.
>
> $~$
>
> Table R1. Inference speed on RTX3090-24GB GPU. Setup: Generate 128 tokens with a batch size of 1.
> | #Bit | Llama-2-7B | Llama-2-70B |
> | --- | --- | --- |
> | 16 | 1.00x (58.1 tokens/second) | 1.00x (5.2 tokens/second)|
> | 2 | 1.29x | 1.17x |
>
>
> $~$
>
> > **Weakness 1. Include more comparisons to recent works, like SqueezeLLM [1] and AdaDim [2].**
>
> Thank you for the suggestion of including more recent baselines.
>
> **Summary: ClusComp outperforms both SqueezeLLM and AdaDim.**
>
> As shown in Table R2, R3 and R4, ClusComp (with a similar level of bit or lower bit) outperforms SqueezeLLM and GPTQ-AdaDim, with a more evident difference for a lower bit. We have added these results to Appendix D.1 in the new submission.
>
> $~$
>
> Table R2. The perplexity on WikiText2. The values in the brackets are the exact bits of ClusComp for different LLMs. The SqueezeLLM results are taken from [1]. Please refer to Table D.1 in the paper for better reading experience.
>
> | Method | #Bit | 2-7B | 2-13B | 2-70B |
> | --- | --- | --- | --- | --- |
> | - | 16 | 5.47 | 4.88 | 3.31 |
> | SqueezeLLM | 4.27 | 5.57 | 4.96 | 3.39 |
> | ClusComp | <= 4.14 | **5.54** (4.14) | **4.94** (4.09)| - |
> | - |
> | SqueezeLLM | 4.05 | 5.62 | 4.99 | 3.41 |
> | ClusComp | 4.03 | - | - | **3.40** (4.03) |
> | - |
> | SqueezeLLM | 3.02 | 6.18 | 5.36 | 3.77 |
> | ClusComp | <=2.89 | **5.86** (2.89) | **5.18** (2.81) | **3.72** (2.72)
> | - |
> | SqueezeLLM | 2.22 | 10.79 | 7.91 | 4.99 |
> | SqueezeLLM | 2.05 | 13.64 | 8.56 | 5.38 |
> | SqueezeLLM | 2.01 | 35.49 | 41.02 | 9.44 |
> | ClusComp | <=2.01 | **7.50** (2.00) | **6.17** (1.99) | **4.83** (1.85) |
>
> Table R3. The accuracy on Llama-2-7B. CSR (commonsense reasoning) here only contains 4 tasks (PIQA, HellaSwag, WinoGrande and ARC-easy) as [2]. The GPTQ-AdaDim results are taken from [2]. Please refer to Table D.2 in the paper for better reading experience.
> | Method | #Bit | MMLU (5-shot) | CSR (0-shot) |
> | --- | --- | --- | --- |
> | - | 16.00 | 46.0 | 67.9 |
> | GPTQ-AdaDim | 4.13 | 45.3 | 67.7 |
> | ClusComp | 4.14 | **45.6** | **68.2** |\
> |-|
> | GPTQ-AdaDim | 3.13 | 41.3 | 66.4 |
> | ClusComp | **2.89** | **43.2** | **67.2** |
>
> Table R4. The accuracy on Llama-2-13B.
> | Method | #Bit | MMLU (5-shot) | CSR (0-shot) |
> | --- | --- | --- | --- |
> | - | 16.00 | 55.6 | 70.3 |
> | GPTQ-AdaDim | 4.13 | 54.6 | 70.1 |
> | ClusComp | 4.09 | **55.1** | **70.8** |
> |-|
> | GPTQ-AdaDim | 3.13 | **52.3** | 68.7 |
> | ClusComp | **2.81** | **52.3** | **69.5** |
>
>
> $~$
>
> > **Weakness 2. Add analysis to explain how the codebooks are distributed to better capture salient weights.**
>
> Really thank you for this suggestion. We have added new figures for visualizing the weight distribution of ClusComp to Appendix D.2 in the new submission.
>
> **Summary: ClusComp’s weight distribution of different bit-levels can better simulate the 16-bit weight distribution than uniform quantization.**
>
> $~$
>
> [1] Kim, et al., "SqueezeLLM: Dense-and-Sparse Quantization", ICML 2024.
>
> [2] Heo, et al., "Rethinking Channel Dimensions to Isolate Outliers for Low-bit Weight Quantization of Large Language Models", ICLR 2024
>
> (1/n)

---

> > ### Author Response · Authors · 2024-11-22
> > **Response n/n**
> >
> > Following response (1/n)
> >
> > $~$
> >
> > > **Question 1. What does divided by a factor mean?**
> >
> > Taking "AQ/4" in Figure 1 as an example, it means that we divide AQ's perplexity by 4. Because AQ's perplexity is too large at the 2-bit level, drawing with this value will significantly stretch the y-axis, making the comparison among other methods less evident. Thank you for this question, we have added an explanation to the caption in the new submission.
> >
> > $~$
> >
> > > **Question 1. What does the triangular marker mean in Figure 1.**
> >
> > Methods with triangular markers use different budgets from the ones with circular markers. The budget here means the number of calibration tokens. Methods with circular markers use about 0.3M calibration tokens, while methods with triangular markers use about 16M tokens. Sorry for the confusion, we have added an explanation to the caption in the new submission.
> >
> > $~$
> >
> > > **Question 2. Simply taking the weight distribution to explain the quantization difficulty of recent LLMs may be insufficient. Activation-aware analysis is required.**
> >
> > Thank you for this suggestion. We have added a new analysis to Appendix D.3 by taking the activation into account. Specifically, we measure the distribution of the Wanda score [3] for Llama-1, Llama-2 and Llama-3 and show the results in Figure D.2.
> >
> > **Summary: Both the weight-only metric (kurtosis on weights) and the weight-activation metric (Wanda score) show that the quantization of Llama series becomes increasingly difficult.**
> >
> > $~$
> >
> > [3] Sun, et al., "A Simple and Effective Pruning Approach for Large Language Models", ICLR 2024
> >
> > $~$
> >
> > ---
> >
> > $~$
> >
> > Thank you for your thoughtful suggestions. We have incorporated the new results in the updated version.
> >
> > If these revisions address your concerns, we kindly request a reconsideration of the scores. Should you have any further questions, we are happy to assist.

---

> > ### Comment · Reviewer_wLyR · 2024-11-25
> > **Thanks for the response**
> >
> > The authors showed an effort to exhaustively answer my questions and concerns, which I appreciate. I will raise my score to borderline accept.
> > I still have some reservations about this work's practical impact as the 17% speedup achieved is marginal, even at 2 bits (which does considerably hurt performance).
> > I am curious about how much headroom there is on kernel optimization.

---

> > > ### Author Response · Authors · 2024-12-01
> > > **100% inference speedup**
> > >
> > > Dear Reviewer wLyR,
> > >
> > > thanks to the extension of the rebuttal deadline, we could finally achieve a **≈ 100% inference speedup** after carefully optimizing the CUDA kernel. Please refer to the [general response](https://openreview.net/forum?id=YZEzVR5awV&noteId=30x15AjS0D) for more details.
> > >
> > > If this work further address your concern, and it is not too much trouble for you, we would really thank you for any update to the score.

---

> ### Author Response · Authors · 2024-11-25
> **Thank you for your increased score**
>
> Dear wLyR,
>
> We are really encouraged by your increasing score. Here we answer your concern about the improvement room for the kernel optimization.
>
> **Summary: Further improvement is possible, according to our finished and in-plan experiments. Currently, we can achieve maximum 40% speedup for 7B model and 22% for 70B model.**
>
> To address the latency issue, we conduct 3 parallel attempts (two are done and one is in progress):
> 1. **(Done)** The results from the first attempt were already shown in our initial repsonse, i.e. **29% speedup over FP16 for a 7B model and 17% for a 70B model**. For this attempt, we mainly fuse all necessary operations (indexing, reshaping, and matrix multiplication).
> 2. **(Done)** Thanks to the suggestion from Reviewer h5Zw who recommends us to **further quantize the codebook from 16-bit (originally) to lower bit**. Therefore. after the block-wise error minimization, we further quantize each cluster in the codebook with symmetric quantization.
>
> From Table R4, we can observe: (1) 8-bit codebook offers the same perplexity as 16-bit’s; (2) 4-bit codebook slightly hurts the performance, but is still comparable to the best baseline at the 4-bit level and significantly outperforms the best baseline at the 2-bit level; (3) The results of the 2-bit codebook are not acceptable.
>
> Since the quantization further reduces the codebook size, we can obtain better inference speed. As shown in Table R5, we obtain **40% speedup for a 7B LLM and 22% for a 70B LLM**.  Notably, our kernel is mainly optimized for the 16-bit codebook. We expect a higher speedup with further optimization. Due to the time limit, we couldn't finish it by now.
>
> 3. **(In progress, fundamental solution)** Before introducing our last attempt, we offer some background knowledge to explain why it is hard to speed up the codebook-based quantization. For faster matrix multiplication, it's better for us to save the codebook in the L1 cache (shared memory) of a thread block that provides fast access to reading. However, the L1 cache in most GPUs is too small to save the codebook. Therefore, we have to save the codebook in the L2 cache (in the previous 2 attempts) that is slower for reading but has a larger storage. Recently, we notice there is a new feature from CUDA, i.e. distributed shared memory. As the terminology implies, we can gather the shared memory from different thread blocks to enlarge the storage. In this way, we can have a larger L1 cache and fast reading access at the same time. We believe this is the fundamental solution to codebook-based quantization method, and might have a great impact to the follow-up research. However, we are still working on it, and haven't obtained the results by now.
>
>  $~$
>
> Table R4. The perplexity of ClusComp with quantized codebook on WikiText2. The best baseline is taken from Table C.1 in the paper. Please refer to Table D.3 in the paper for better reading experience and more results (70B LLM offers the same observation).
> | Method | Bit for codebook | Avg. Bit | Llama-2-7B | Llama-3-8B |
> | --- | --- | --- | --- | --- |
> | - | - | 16.00 | 5.47 | 6.12 |
> | Best baseline | - | 4.13 | 5.58 | 6.50 |
> | ClusComp | 16 | 4.14 | 5.54 | 6.39 |
> | ClusComp | 8 | 4.11 | 5.54 | 6.39 |
> | ClusComp | 4 | 4.07 | 5.59 | 6.52 |
> | ClusComp | 2 | 4.05 | 25.44 | 1.2e5 |
> |-|
> | Best baseline | - | 2.00 | 35.07 | 85.10 |
> | ClusComp | 16 | 2.01 | 7.50 | 12.33 |
> | ClusComp | 8 | 1.92 | 7.50 | 12.33 |
> | ClusComp | 4 | 1.86 | 7.63 | 12.77 |
> | ClusComp | 2 | 1.83 | 6.5e3 | 1.7e5 |
>
> Table R5. Inference speed on RTX3090-24GB GPU.
> | Bit for codebook | Avg. Bit | Llama-2-7B | Llama-2-70B |
> | --- | --- | --- | --- |
> | - | 16 | 1.00x | 1.00x |
> | 16 | 2.01 | 1.29x | 1.17x |
> | 8 | 1.92 | 1.31x | 1.19x |
> | 4 | 1.86 | 1.40x | 1.22x |

---

### Author Response · Authors · 2024-11-22
**New results summarization**

Here we summarize the new results in the updated submission, you can selectively read them if you are interested. We highlight some for your easy choice.

| Table or Figure | Content | Where for details (i.e. response to which point of which reviewer) |
| --- | --- | --- |
| Table D.1 and D.2 | Compare to SqueezeLLM and AdaDim | Weakness 1 of Reviewer wLyR |
| Figure D.1 | **Visualization of ClusComp's weight distribution** | Weakness 2 of Reviewer wLyR |
| Figure D.2 | **More evidence to show the increasing quantization difficulty of Llama series** | Question 2 of Reviewer wLyR |
| Table D.3 | Further quantization of the codebook | Question 1 of Reviewer h5Zw |
| Table D.4 | **Ablation study on the cluster number and cluster dimension** | Question 3 of Reviewer h5Zw |
| Figure D.3 | Add GPTQ, SqueezeLLM and QuIP to Figure 1 | Strength 2 of  Reviewer dutm |

---

> ### Author Response · Authors · 2024-12-01
> **100% inference speedup**
>
> Thanks to the extension of rebuttal deadline. **Our currently optimized CUDA kernel for ClusComp can achieve ~100% speedup over the FP16 model**.
>
> **Experimental setting**: We further quantize the codebook (originally in 16-bit) to lower bit (4-bit) after block-wise error minimization (suggested by Reviewer h5Zw). Such a design further reduces the codebook size with minimal performance degradation, therefore leading to higher speedup with a carefully optimized kernel. We measure the time required for generating 128 tokens with a batch size of 1 (2048 tokens offer a similar result).
>
> **Perplexity performance**: As shown in Table OR1, we can observe: (1) 8-bit codebook offers the same perplexity as 16-bit’s; (2) **4-bit codebook slightly hurts the performance, but still achieveing the same perplexity to the best baseline for 4-bit LLMs and significantly outperforming the best baseline for 2-bit LLMs.** (3) The results of the 2-bit codebook are not acceptable.
>
> **Inference speedup**: As shown in Table OR2, **the inference speedup is 110% for a 7B LLM, and 81% for a 70B LLM.**
>
> $~$
>
> Table OR1. The perplexity of ClusComp with quantized codebook on WikiText2. The best baseline is taken from Table C.1 in the paper. Please refer to Table D.3 in the paper for better reading experience and more results (70B LLM offers the same observation).
> | Method | Bit for codebook | Avg. Bit | Llama-2-7B | Llama-3-8B |
> | --- | --- | --- | --- | --- |
> | - | - | 16.00 | 5.47 | 6.12 |
> | Best baseline | - | 4.13 | 5.58 | 6.50 |
> | ClusComp | 16 | 4.14 | 5.54 | 6.39 |
> | ClusComp | 8 | 4.11 | 5.54 | 6.39 |
> | ClusComp | 4 | 4.07 | 5.59 | 6.52 |
> | ClusComp | 2 | 4.05 | 25.44 | 1.2e5 |
> |-|
> | Best baseline | - | 2.00 | 35.07 | 85.10 |
> | ClusComp | 16 | 2.01 | 7.50 | 12.33 |
> | ClusComp | 8 | 1.92 | 7.50 | 12.33 |
> | ClusComp | 4 | 1.86 | 7.63 | 12.77 |
> | ClusComp | 2 | 1.83 | 6.5e3 | 1.7e5 |
>
> Table OR2. Inference speed on a A6000-48GB GPU.
> | Bit for codebook | Avg. Bit | Llama-2-7B | Llama-2-70B |
> | --- | --- | --- | --- |
> | - | 16 | 1.00x | 1.00x |
> | 4 | 1.86 | 2.10x | 1.81x |

---

### Meta-Review · Area_Chair_Zp2B · 2024-12-20

**Metareview:**

This paper presents ClusComp, a novel model compression technique for large language models (LLMs) that leverages clustering-based quantization to achieve ultra-low bit-widths, down to 1-bit precision. The authors demonstrate that ClusComp can significantly reduce model size while maintaining performance, showing promising results across various tasks and model sizes. The method is particularly effective in the 2-4 bit range, and the authors claim it outperforms existing quantization techniques while enabling efficient finetuning.

However, despite the authors' efforts to address the concerns raised by reviewers, several critical issues remain unresolved. One reviewer noted that, at bit-widths of 3 or higher, the throughput is significantly reduced without yielding corresponding improvements in accuracy, making ClusComp inferior to other methods in these settings. The reviewer also pointed out that while the 1-bit compression model is promising, the trade-offs in throughput and performance degradation need further exploration. Despite the authors' attempts to address these concerns, the reviewer did not find the responses convincing, particularly regarding the practical impact and optimization of throughput for the 1-bit model.

The Area Chair has carefully considered the reviewers' feedback. While the authors made efforts to respond to the concerns and clarify their methodology, the practical impact of ClusComp remains limited, especially at bit-widths above 2, where performance degradation becomes more significant. Given the competitive nature of the conference and the need for further refinement, particularly around kernel optimization and the throughput-performance trade-off, the decision is to reject the submission. While the contribution is interesting and shows potential, it requires more thorough optimization and evaluation to demonstrate its real-world applicability and impact in model compression for large-scale LLMs.

**Additional Comments On Reviewer Discussion:**

During the rebuttal period, the authors made efforts to address several concerns raised by the reviewers. One reviewer pointed out that ClusComp shows reduced throughput and similar accuracy at bit-widths greater than 2, making it less advantageous compared to existing methods. The authors responded by clarifying that the method is optimized for ultra-low bit-widths (particularly 1-bit), where it shows the most promise. However, the reviewer remained unconvinced, noting that the practical impact of the method, especially for models with bit-widths of 3 or higher, still seemed marginal.

Another reviewer expressed interest in understanding the kernel optimizations for the 1-bit model and asked for more detailed benchmarking to assess throughput trade-offs. The authors provided some additional clarification on their kernel design and attempted to justify the throughput hit. However, the reviewer did not find this sufficiently convincing and suggested that the paper would benefit from a more focused evaluation of the 1-bit model and a clearer understanding of the trade-offs in real-world use cases.

In weighing these points, the Area Chair found that while the authors made valid efforts to address the concerns, the core issues regarding the practical applicability of the method, particularly at bit-widths above 2 and the throughput trade-offs, were not fully resolved. Therefore, the decision to reject was made, given that the method’s real-world impact remains uncertain, especially in the competitive context of the conference.

---

### Decision · Program_Chairs · 2025-01-22

Reject